# PageRank Bandits for Link Prediction

**Yikun Ban**[1*]**, Jiaru Zou**[1*]**, Zihao Li**[1]**, Yunzhe Qi**[1]**, Dongqi Fu**[2]**, Jian Kang**[3]**,**
**Hanghang Tong**[1]**, Jingrui He**[1]
[1]University of Illinois Urbana-Champaign, [2]Meta AI, [3]University of Rochester
[1]{yikunb2, jiaruz2, zihaoli5, yunzheq2, htong, jingrui}@illinois.edu
[2]dongqifu@meta.com, [3]jian.kang@rochester.edu

## Abstract

Link prediction is a critical problem in graph learning with broad applications such as recommender systems and knowledge graph completion. Numerous research efforts have been directed at solving this problem, including approaches based on similarity metrics and Graph Neural Networks (GNN). However, most existing solutions are still rooted in conventional supervised learning, which makes it challenging to adapt over time to changing customer interests and to address the inherent dilemma of exploitation versus exploration in link prediction. To tackle these challenges, this paper reformulates link prediction as a sequential decision-making process, where each link prediction interaction occurs sequentially. We propose a novel fusion algorithm, PRB (PageRank Bandits), which is the first to combine contextual bandits with PageRank for collaborative exploitation and exploration. We also introduce a new reward formulation and provide a theoretical performance guarantee for PRB. Finally, we extensively evaluate PRB in both online and offline settings, comparing it with bandit-based and graph-based methods. The empirical success of PRB demonstrates the value of the proposed fusion approach. Our code is released at https://github.com/jiaruzouu/PRB

## 1 Introduction

Link prediction is an essential problem in graph machine learning, focusing on predicting whether a link will exist between two nodes. Given the ubiquitous graph data in real-world applications, link prediction has become a powerful tool in domains such as recommender systems [72] and knowledge graph completion [49, 41]. Considerable research efforts have been dedicated to solving this problem. One type of classic research approaches is heuristic-based methods, which infer the likelihood of links based on node similarity metrics [43, 46]. Graph Neural Networks (GNNs) have been widely utilized for link prediction. For example, Graph Autoencoders leverage Message Passing Neural Network (MPNN) representations to predict links [29]. Recently, MPNNs have been combined with structural features to better explore pairwise relations between target nodes [73, 70, 18, 61].

Existing supervised-learning-based methods for link prediction are designed for either the static [73, 70, 18, 61] or relatively dynamic environment [64, 55, 62, 58, 69, 19, 27, 26, 75], they (chronologically) split the dataset into training and testing sets. Due to the dynamic and evolving nature of many real-world graphs, ideal link prediction methods should adapt over time to consistently meet the contexts and goals of the serving nodes. For instance, in short-video recommender systems, both video content and user preferences change dynamically over time [28]. Another significant challenge is the dilemma of exploitation and exploration in link prediction. The learner must not only exploit past collected data to predict links with high likelihood but also explore lower-confidence target nodes to acquire new knowledge for long-term benefits. For example, in social recommendations, it

---

*Equal contribution.

is necessary to prioritize popular users by 'exploiting' knowledge gained from previous interactions, while also'exploring' potential value from new or under-explored users to seek long-term benefits [7]. Furthermore, while existing works often analyze time and space complexity, they generally lack theoretical guarantees regarding the performance of link prediction. To address these challenges, in this paper, we make the following contributions:

**Problem Formulation and Algorithm**. We formulate the task of link prediction as sequential decision-making under the framework of contextual bandits, where each interaction of link prediction is regarded as one round of decision-making. We introduce a pseudo-regret metric to evaluate the performance of this decision process. More specifically, we propose a fusion algorithm named PRB (PageRank Bandits), which combines the exploitation and exploration balance of contextual bandits with the graph structure utilization of PageRank [59, 42]. Compared to contextual bandit approaches, PRB leverages graph connectivity for an aggregated representation. In contrast to PageRank, it incorporates the principles of exploitation and exploration from contextual bandits to achieve a collaborative trade-off. Additionally, we extend PRB to node classification by introducing a novel transformation from node classification to link prediction, thereby broadening the applicability of PRB.

**Theoretical Analysis**. We introduce a new formulation of the reward function to represent the mapping from both node contexts and graph connectivity to the reward. We provide one theoretical guarantee for the link prediction performance of the proposed algorithm, demonstrating that the cumulative regret induced by PRB can grow sub-linearly with respect to the number of rounds. This regret upper bound also provides insights into the relationship between the reward and damping factor, as well as the required realization complexity of the neural function class.

**Empirical Evaluation**. We extensively evaluate PRB in two mainstream settings. (1) Online Link Prediction. In this setting, each link prediction is made sequentially. In each round, given a serving node, the model is required to choose one target node that has the highest likelihood of forming a link with the serving node. The model then observes feedback and performs corresponding optimizations. The goal is to minimize regret over $T$ rounds (e.g., $T = 10,000$). We compare PRB with state-of-the-art (SOTA) bandit-based approaches (e.g., [76, 12]), which are designed for sequential decision-making. PRB significantly outperforms these bandit-based baselines, demonstrating the success of fusing contextual bandits with PageRank for collaborative exploitation and exploration. (2) Offline Link Prediction. In this setting, both training and testing data are provided, following the typical supervised learning process. Although PRB is designed for online learning, it can be directly applied to offline learning on the training data. We then use the trained model to perform link prediction on the testing data, comparing it with SOTA GNNs-based methods (e.g., [18, 61]). The superior performance of PRB indicates that principled exploitation and exploration can break the performance bottleneck in link prediction. Additionally, we conduct ablation and sensitivity studies for a comprehensive evaluation of PRB.

## 2  Related Work

**Contextual Bandits**. The first line of works studies the linear reward assumption, typically calculated using ridge regression [39, 8, 1, 60, 21, 53]. Linear UCB-based bandit algorithms [1, 9, 40] and linear Thompson Sampling [4, 2] can achieve satisfactory performance and a near-optimal regret bound of $\tilde{\mathcal{O}}(\sqrt{T})$. To learn general reward functions, deep neural networks have been adapted to bandits in various ways [10, 11]. [54, 47] develop $L$-layer DNNs to learn arm embeddings and apply Thompson Sampling on the final layer for exploration. [76] introduced the first provable neural-based contextual bandit algorithm with a UCB exploration strategy, and [74] later extended to the TS framework. [22] provides sharper regret upper bound for neural bandits with neural online regression. Their regret analysis builds on recent advances in the convergence theory of over-parameterized neural networks [24, 5] and uses the Neural Tangent Kernel [34, 6] to establish connections with linear contextual bandits [1]. [12, 13] retains the powerful representation ability of neural networks to learn the reward function while using another neural network for exploration. [52, 51] integrates exploitation-exploration neural networks into the graph neural networks for fine-grained exploration and exploration. Recently, neural bandits have been adapted to solve other learning problems, such as active learning[14, 7], meta learning[53].

**Link Prediction Models.** Three primary approaches have been identified for link prediction models. Node embedding methods, as described by previous work [50, 30, 57, 23, 44, 45, 25], focus on

mapping each node to an embedding vector and leveraging these embeddings to predict connections. Another approach involves link prediction heuristics, as explored by [43, 15, 3, 77], which utilize crafted structural features and network topology to estimate the likelihood of connections between nodes in a network. The third category employs GNNs for predicting link existence; notable is the Graph Autoencoder (GAE) [36], which learns low-dimensional representations of graph-structured data through an unsupervised learning process. GAE utilizes the inner product of MPNN representations of target nodes to forecast links but might not capture pairwise relations between nodes effectively. More sophisticated GNN models that combine MPNN with additional structural features, such as those by [71, 70, 18], have demonstrated superior performance by integrating both node and structural attributes. One such combined architecture is SF-then-MPNN, as adopted by [71, 78]. In this approach, the input graph is first enriched with structural features (SF) and then processed by the MPNN to enhance its expressivity. However, since structural features change with each target link, the MPNN must be re-run for each link, reducing scalability. For instance, the SEAL model [71] first enhances node features by incorporating the shortest path distances and extracting k-hop subgraphs, then applies MPNN across these subgraphs to generate more comprehensive link representations. Another combined architecture is SF-and-MPNN. Models like Neo-GNN [70] and BUDDY [18] apply MPNN to the entire graph and concatenate features such as common neighbor counts to enhance representational fidelity. In addition, [61] has developed the Neural Common Neighbor with Completion (NCNC) which utilizes the MPNN-then-SF architecture to achieve higher expressivity and address the graph incompleteness.

Recently, representation learning on temporal graphs for link prediction has also been widely studied to exploit patterns in historical sequences, particularly with GNN-based methods [58, 69, 19, 64, 62, 55]. However, these approaches are still conventional supervised-learning-based methods that chronologically split the dataset into training and testing sets. Specifically, these methods train a GNN-based model on the temporal training data and then employ the trained model to predict links in the test data. In contrast, we formulate link predictions as sequential decision-making, where each link prediction is made sequentially. Node classification[16, 67, 66] is also a prominent direction in graph learning, but it is not the main focus of this paper.

## 3 Problem Definition

Let $G_0 = (V, E_0)$ be an undirected graph at initialization, where $V$ is the set of $n$ nodes, $|V| = n$, and $E_0 \subseteq V \times V$ represents the set of edges. $E_0$ can be an empty set in the cold-start setting or include some existing edges with a warm start. Each node $v_i \in V$ is associated with a context vector $x_{0,i} \in \mathbb{R}^d$. Then, we formulate link prediction as the problem of sequential decision-making under the framework of contextual bandits. Suppose the learner is required to finish a total of $T$ link predictions. We adapt the above notation to all the evolving $T$ graphs $\{G_t = (V, E_t)\}_{t=0}^{T-1}$ and let $[T] = \{1, \ldots, T\}$. In a round of link prediction $t \in [T]$, given $G_{t-1} = (V, E_{t-1})$, the learner is presented with a serving node $v_t \in V$ and a set of $k$ candidate nodes $\mathcal{V}_t = \{v_{t,1}, \ldots, v_{t,k}\} \subseteq V$, where $\mathcal{V}_t$ is associated with the corresponding $k$ contexts $\mathcal{X}_t = \{x_{t,1}, \ldots, x_{t,k}\}$ and $|\mathcal{V}_t| = k$. In the scenario of social recommendation, $v_t$ can be considered as the user that the platform (learner) intends to recommend potential friends to, and the other candidate users will be represented by $\mathcal{V}_t$. $\mathcal{V}_t$ can be set as the remaining nodes $\mathcal{V}_t = V_t/v_t$ or formed by some pre-selection algorithm $\mathcal{V}_t \subset V_t$.

The goal of the learner is to predict which node in $\mathcal{V}_t$ will generate a link or edge with $v_t$. Therefore, we can consider each node in $\mathcal{V}_t$ as an arm, and aim to select the arm with the maximal reward or the arm with the maximal probability of generating an edge with $v_t$. For simplicity, we define the reward of link prediction as the binary reward. Let $v_{t,\hat{i}} \in \mathcal{V}_t$ be the node selected by the learner. Then, the corresponding reward is defined as $r_{t,\hat{i}} = 1$ if the link $[v_t, v_{t,\hat{i}}]$ is really generated; otherwise, $r_{t,\hat{i}} = 0$. After observing the reward $r_{t,\hat{i}}$, we update $E_{t-1}$ to obtain the new edge set $E_t$, and thus new $G_t$.

For any node $v_{t,i} \in \mathcal{V}_t$, denote by $\mathcal{D}_{\mathcal{Y}|x_{t,i}}$ the conditional distribution of the random reward $r_{t,i}$ with respect to $x_{t,i}$, where $\mathcal{Y} = \{1, 0\}$. Then, inspired by the literature of contextual bandits, we define the following *pseudo* regret:

$$\mathbf{R}_T = \sum_{t=1}^T \left( \mathbb{E}_{r_{t,i^*} \sim \mathcal{D}_{\mathcal{Y}|x_{t,i^*}}}[r_{t,i^*}] - \mathbb{E}_{r_{t,\hat{i}} \sim \mathcal{D}_{\mathcal{Y}|x_{t,\hat{i}}}}[r_{t,\hat{i}}] \right) = \mathbb{P}(r_{t,i^*} = 1|x_{t,i^*}) - \mathbb{P}(r_{t,\hat{i}} = 1|x_{t,\hat{i}})$$

$$(3.1)$$

**Algorithm 1** PRB (PageRank Bandits)

---

**Input:** $f_1, f_2, T, G_0, \eta_1, \eta_2$ (learning rate), $\alpha$ (damping factor)
1: Initialize $\theta_0^1, \theta_0^2$
2: **for** $t = 1, 2, \ldots, T$ **do**
3:      Observe serving node $v_t$, candidate nodes $\mathcal{V}_t$, contexts $\mathcal{X}_t$ and Graph $G_{t-1}$
4:      $\mathbf{h}_t = \mathbf{0}$
5:      **for** each $v_{t,i} \in \mathcal{V}_t$ **do**
6:          $\mathbf{h}_t[i] = f_1\left(x_{t,i}; \theta_{t-1}^1\right) + f_2\left(\phi\left(x_{t,i}\right); \theta_{t-1}^2\right)$
7:      **end for**
8:      Compute $\mathbf{P}_t$ based on $G_{t-1}$
9:      Solve $\mathbf{v}_t = \alpha \mathbf{P}_t \mathbf{v}_t + (1-\alpha)\mathbf{h}_t$
10:     Select $\hat{i} = \arg\max_{v_{t,i} \in \mathcal{V}_t} \mathbf{v}_t[i]$
11:     Observe $r_{t,\hat{i}}$
12:     **if** $r_{t,\hat{i}} == 1$ **then**
13:        Add $[v_t, v_{t,\hat{i}}]$ to $G_{t-1}$ and set as $G_t$
14:     **else**
15:        $G_t = G_{t-1}$
16:     **end if**
17:     $\theta_t^1 = \theta_{t-1}^1 - \eta_1 \nabla_{\theta_{t-1}^1} \mathcal{L}\left(x_{t,\hat{i}}, r_{t,\hat{i}}; \theta_{t-1}^1\right)$
18:     $\theta_t^2 = \theta_{t-1}^2 - \eta_2 \nabla_{\theta_{t-1}^2} \mathcal{L}\left(\phi(x_{t,\hat{i}}), r_{t,\hat{i}} - f_1(x_{t,\hat{i}}; \theta_{t-1}^1); \theta_{t-1}^2\right)$
19: **end for**

---

where $i^* = \arg\max_{v_{t,i} \in \mathcal{V}_t} \mathbb{P}(r_{t,i} = 1 | x_{t,i})$, the tie is broken randomly, and $\hat{i}$ is the index of selected node. $\mathbf{R}_T$ reflects the performance difference of the learned model from the Bayes-optimal predictor. The goal of the learner is to minimize $\mathbf{R}_T$.

## 4 Proposed Algorithms

Algorithm 1 describes the proposed algorithm PRB. It integrates contextual bandits and PageRank to combine the power of balancing exploitation and exploration with graph connectivity. The first step is to balance the exploitation and exploration in terms of the reward mapping concerning node contexts, and the second step is to propagate the exploitation and exploration score via graph connectivity.

To exploit the node contexts, we use a neural network to estimate rewards from the node contexts. Let $f_1(\cdot; \theta^1)$ be a neural network to learn the mapping from the node context to the reward. Denote the initialized parameter of $f_1$ by $\theta_0^1$. In round $t$, let $\theta_{t-1}^1$ be parameter trained on the collected data of previous $t-1$ rounds including all selected nodes and the received rewards. Given the serving node $v_t$, for any candidate node $v_{t,i} \in \mathcal{V}_t$, $f_1(x_{t,i}; \theta_{t-1}^1), i \in \mathcal{V}_t$ is the estimated reward by greedily exploiting the observed contexts, which we refer to as "exploitation". Suppose $\hat{i}$ is the index of selected nodes. To update $\theta_{t-1}^1$, we can conduct stochastic gradient descent to update $\theta^1$ based on the collected training sample $(x_{t,\hat{i}}, r_{t,\hat{i}})$ with the squared loss function $\mathcal{L}\left(x_{t,\hat{i}}, r_{t,\hat{i}}; \theta_{t-1}^1\right) = [f(x_{t,\hat{i}}; \theta_{t-1}^1) - r_{t,\hat{i}}]^2/2$. Denote the updated parameters by $\theta_t^1$ for the next round of link prediction.

In addition to exploiting the observed contexts, we employ another neural network to estimate the potential gain of each candidate node in terms of reward for exploration. This idea is inspired by [12]. Denote the exploration network by $f_2(\cdot; \theta^2)$. $f_2$ is to learn the mapping from node contexts and the discriminative ability of $f_1$ to the potential gain. In round $t \in [T]$, given node context $x_{t,i} \in \mathcal{V}_t$ and its estimated reward $f_1(x_{t,i}; \theta_{t-1}^1)$, the input of $f_2$ is the gradient of $f_1(x_{t,i}; \theta_{t-1}^1)$ with respect to $\theta_{t-1}^1$, denoted by $\phi(x_{t,i})$, and $f_2(\phi(x_{t,i}); \theta_{t-1}^2)$ is the estimated potential gain. After the learner selects the node $x_{t,\hat{i}}$ and observes the reward $r_{t,\hat{i}}$, the potential gain is defined as $r_{t,\hat{i}} - f_1(x_{t,i}; \theta_{t-1}^1)$, which is used to train $f_2$. Thus, after this interaction, we conduct the stochastic gradient descent to update $\theta^2$ based on the collected sample $(\phi(x_{t,\hat{i}}), r_{t,\hat{i}} - f_1(x_{t,i}; \theta_{t-1}^1))$ with the squared loss function $\mathcal{L}\left(\phi(x_{t,\hat{i}}), r_{t,\hat{i}} - f_1(x_{t,\hat{i}}; \theta_{t-1}^1); \theta_{t-1}^2\right) = [f(\phi_{t,i}; \theta_{t-1}^2) - (r_{t,\hat{i}} - f_1(x_{t,i}; \theta_{t-1}^1))]^2/2$. Denote by $\theta_t^2$

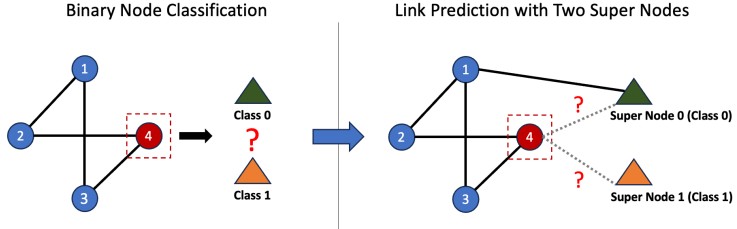

Figure 1: **Transforming Node Classification to Link Prediction**. Consider a binary node classification problem. In the left figure, given a graph, the learner tries to classify node 4 into one of two classes. First, we add two supernodes to the graph, each representing one of the classes. The node classification problem is then transformed into predicting links between node 4 and the two supernodes in the right figure. Suppose the learner predicts that a link will exist between node 4 and supernode 0. If node 4 belongs to Class 0, the reward is 1, and an edge is added between node 4 and supernode 0; otherwise, the reward is 0, and an edge is added between node 4 and supernode 1.

the updated parameters of $f_2$ for the next round of link prediction. The reasons for setting $\phi(x_{t,i})$ as the input of $f_2$ are as follows: (1) it incorporates the information of both $x_{t,\hat{i}}$ and discriminative ability of $f_1(\cdot; \theta^1_{t-1})$; (2) the statistical form of the confidence interval for reward estimation can be regarded as the mapping function from $\phi(x_{t,i})$ to the potential gain, and $f_2$ is to learn the unknown mapping [12].

The previous steps demonstrate the exploitation and exploration of node contexts to facilitate decision-making in link prediction. Since graph connectivity is also crucial, we next introduce our method of integrating the bandit principle with PageRank to enable collaborative exploitation and exploration. PageRank calculates the stationary distribution of the random walker starting from some node, iteratively moving to a random neighbor with probability $\alpha$ (damping factor) or returning to its original position with probability $1 - \alpha$. Let $\mathbf{v}_t$ be the stationary distribution vector calculated based on the graph $G_t$. Then, $\mathbf{v}_t$ satisfies:

$$\mathbf{v}_t = \alpha \mathbf{P}_t \mathbf{v}_t + (1 - \alpha) \mathbf{h}_t \tag{4.1}$$

where $\mathbf{P}_t \in \mathbb{E}^{n \times n}$ is the transition matrix built on $G_{t-1}$ and $\mathbf{h}_t$ is typically regarded as a position vector to mark the starting node. $\mathbf{P}_t$ is computed as $\mathbf{D}^{-1}_{t-1} \mathbf{A}_{t-1}$, where $\mathbf{D}_{t-1} \in \mathbb{R}^{n \times n}$ is the degree matrix of $G_{t-1}$ and $\mathbf{A}_{t-1} \in \mathbb{R}^{n \times n}$ is the adjacency matrix of $G_{t-1}$.

Here we propose to use $\mathbf{h}_t$ to include the starting exploitation and exploration scores of candidate nodes, defined as:

$$i \in \mathcal{V}_t, \mathbf{h}_t[i] = f_1(x_{t,i}; \theta^1_{t-1}) + f_2(x_{t,i}; \theta^2_{t-1}), \text{ and } i \in V/\mathcal{V}_t, \mathbf{h}_t[i] = 0. \tag{4.2}$$

Therefore, $\mathbf{v}_t$ is the vector for the final decision-making based on collaborative exploitation and exploration. Some research efforts have been devoted to accelerating the calculation of Eq.4.1 in the evolving graph, e.g., [42], which can be integrated into PRB (Line 9 in Algorithm 1) to boost its efficiency and scalability.

**PRB for Node Classification**. We also extend PRB to solve the problem of node classification as illustrated in Figure 1. Consider a $k$-class classification problem. We add $k$ super nodes $\{\tilde{v}_1, \tilde{v}_2, \ldots, \tilde{v}_k\}$ to the graph, which represents $k$ classes, respectively. Then, we transform the node classification problem into the link prediction problem, aiming to predict the link between the serving node and the $k$ super nodes. To be specific, in round $t \in [T]$, the learner is presented with the serving node $v_t$ and the $k$ candidate (super) nodes $\mathcal{V}_t = \{\tilde{v}_1, \tilde{v}_2, \ldots, \tilde{v}_k\}$ associated with $k$ corresponding contexts $\mathcal{X}_t = \{x_{t,1}, x_{t,2}, \ldots, x_{t,k}\}$. Recall $x_t$ is the context associated with $v_t$. Then, we define the contexts of super nodes as $x_{t,1} = [x_t^\top, \mathbf{0}, \ldots, \mathbf{0}]^\top, x_{t,2} = [\mathbf{0}, x_t^\top, \ldots, \mathbf{0}]^\top, \ldots, x_{t,k} = [\mathbf{0}, \mathbf{0}, \ldots, x_t]^\top$, $x_{t,i} \in \mathbb{R}^{kd}, i \in [k]$. This context definition is adopted from neural contextual bandits [12, 76]. Then, the learner is required to select one node from $\mathcal{V}_t$. Let $\tilde{v}_{i_t}$ be the selected node and $\tilde{v}_{i_t^*}$ be ground-truth node ($i_t^*$ is the index of ground-truth class of node $v_t$). Then, after observing the reward $r_{t,i_t}$, one edge $[v_t, \tilde{v}_{i_t}]$ is added to the graph $G_{t-1}$, if $v_t$ belongs to the class $i_t$, i.e., $i_t = i_t^*$ and reward $r_{t,i_t} = 1$. Otherwise, $r_{t,i_t} = 0$ and the edge $[v_t, \tilde{v}_{i_t^*}]$ is added to $G_{t-1}$. Then, we can naturally apply PRB to this problem. We detail our extended algorithm for node classification in Algorithm 2.

**PRB Greedy.** We also introduce a greedy version of PRB which integrates PageRank solely with contextual bandit exploitation, as outlined in Algorithm 3. We will compare each variant of algorithms in our experiment section.

# 5  Regret Analysis

In this section, we provide the theoretical analysis of PRB by bounding the regret defined in Eq.3.1. One important step is the definition of the reward function, as this problem is different from the standard bandit setting that focuses on the arm (node) contexts and does not take into account the graph connectivity. First, we define the following general function to represent the mapping from the node contexts to the reward. Given the serving node $v_t$ and an arm node $v_{t,i} \in V_t$ associated with the context $x_{t,i}$, the reward conditioned on $v_t$ and $v_{t,i}$ is assumed to be governed by the function:

$$\mathbb{E}[\tilde{r}_{t,i}|v_t, v_{t,i}] = y(x_{t,i}) \tag{5.1}$$

where $y$ is an unknown but bounded function that can be either linear or non-linear. Next, we provide the formulation of the final reward function. In round $t \in [T]$, let $\mathbf{y}_t$ be the vector to represent the expected rewards of all candidate arms $\mathbf{y}_t = [y(x_{t,i}) : v_{t,i} \in V_t]$. Given the graph $G_{t-1}$, its normalized adjacency matrix $\mathbf{P}_t$, and the damping factor $\alpha$, inspired by PageRank, the optimizing problem is defined as: $\mathbf{v}_t^* = \arg\min_{\mathbf{v}} \alpha \mathbf{v}^\top (\mathbf{I} - \mathbf{P}_t)\mathbf{v} + (1 - \alpha)\|\mathbf{v} - \mathbf{y}_t\|_2^2/2$. Then, its optimal solution is

$$\mathbf{v}_t^* = \alpha \mathbf{P}_t \mathbf{v}_t^* + (1 - \alpha)\mathbf{y}_t. \tag{5.2}$$

For any candidate node $v_{t,i} \in \mathcal{V}_t$, we define its expected reward as $\mathbb{E}_{r_{t,i} \sim \mathcal{D}_{\mathcal{Y}|x_{t,i}}}[r_{t,i}] = \mathbf{v}_t^*[i]$. $\mathbf{v}_t^*$ is a flexible reward function that reflects the mapping relation of both node contexts and graph connectivity. $\alpha$ is a hyper-parameter to trade-off between the leading role of graph connectivity and node contexts. When $\alpha = 0$, $\mathbf{v}_t^*$ turns into the reward function in contextual bandits [76, 12]; when $\alpha = 1$, $\mathbf{v}_t^*$ is the optimal solution solely for graph connectivity. Here, we assume $\alpha$ is a prior knowledge. Finally, the pseudo-regret is defined as

$$\mathbf{R}_T = \sum_{t=1}^{T} \left( \mathbf{v}_t^*[i^*] - \mathbf{v}_t^*[\hat{i}] \right). \tag{5.3}$$

where $i^* = \arg\max_{v_{t,i} \in \mathcal{V}_t} \mathbf{v}_t^*[i]$ and $\hat{i}$ is the index of the selected node. The regret analysis is associated with the Neural Tangent Kernel (NTK) matrix as follows:

**Definition 5.1** (NTK [34, 63]). *Let $\mathcal{N}$ denote the normal distribution. Given all data instances $\{x_t\}_{t=1}^{Tk}$, for $i, j \in [Tk]$, define*

$$\mathbf{H}_{i,j}^0 = \Sigma_{i,j}^0 = \langle x_i, x_j \rangle, \quad \mathbf{A}_{i,j}^l = \begin{pmatrix} \Sigma_{i,i}^l & \Sigma_{i,j}^l \\ \Sigma_{j,i}^l & \Sigma_{j,j}^l \end{pmatrix}$$

$$\Sigma_{i,j}^l = 2\mathbb{E}_{a,b \sim \mathcal{N}(\mathbf{0}, \mathbf{A}_{i,j}^{l-1})}[\sigma(a)\sigma(b)],$$

$$\mathbf{H}_{i,j}^l = 2\mathbf{H}_{i,j}^{l-1}\mathbb{E}_{a,b \sim \mathcal{N}(\mathbf{0}, \mathbf{A}_{i,j}^{l-1})}[\sigma'(a)\sigma'(b)] + \Sigma_{i,j}^l.$$

*Then, the NTK matrix is defined as $\mathbf{H} = (\mathbf{H}^L + \Sigma^L)/2$.*

**Assumption 5.1.** There exists $\lambda_0 > 0$, such that $\mathbf{H} \succeq \lambda_0 \mathbf{I}$.

The assumption 5.1 is generally made in the literature of neural bandits [76, 74, 20, 35, 12, 10, 65] to ensure the existence of a solution for NTK regression.

As the standard setting in contextual bandits, all node contexts are normalized to the unit length. Given $x_{t,i} \in \mathbb{R}^d$ with $\|x_{t,i}\|_2 = 1$, $t \in [T]$, $i \in [k]$, without loss of generality, we define a fully-connected network with depth $L \geq 2$ and width $m$:

$$f(x_{t,i}; \theta) = \mathbf{W}_L \sigma(\mathbf{W}_{L-1} \sigma(\mathbf{W}_{L-2} \ldots \sigma(\mathbf{W}_1 x_{t,i}))) \tag{5.4}$$

where $\sigma$ is the ReLU activation function, $\mathbf{W}_1 \in \mathbb{R}^{m \times d}$, $\mathbf{W}_l \in \mathbb{R}^{m \times m}$, for $2 \leq l \leq L - 1$, $\mathbf{W}^L \in \mathbb{R}^{1 \times m}$, and $\theta = [\text{vec}(\mathbf{W}_1)^\top, \text{vec}(\mathbf{W}_2)^\top, \ldots, \text{vec}(\mathbf{W}_L)^\top]^\top \in \mathbb{R}^p$. Note that our analysis can also be readily generalized to other neural architectures such as CNNs and ResNet [5, 24]. We employ the following initialization [17] for $\theta$: For $l \in [L - 1]$, each entry of $\mathbf{W}_l$ is drawn from the normal distribution $\mathcal{N}(0, 2/m)$; each entry of $\mathbf{W}_L$ is drawn from the normal distribution $\mathcal{N}(0, 1/m)$. The network $f_1$ and $f_2$ follows the structure of $f$. Define $\mathbf{y} = [y(x_{t,i}) : t \in [T], i \in [k]]$. Finally, we provide the performance guarantee as stated in the following Theorem.

**Theorem 5.1.** *Given the number of rounds $T$, for any $\alpha, \delta \in (0, 1)$, suppose $m \geq \widetilde{\Omega}(poly(T, L) \cdot k \log(1/\delta))$, $\eta_1 = \eta_2 = \frac{T^3}{\sqrt{m}}$ and set $\tilde{r}_{t,i} = r_{t,i}, t \in [T], i \in [k]$. Then, with probability at least $1 - \delta$ over the initialization, Algorithm 1 achieves the following regret upper bound:*

$$\mathbf{R}_T \leq \widetilde{\mathcal{O}}(\sqrt{\tilde{d}kT}) \cdot \sqrt{\max(\tilde{d}, S^2)} \tag{5.5}$$

*where $\widetilde{d} = \frac{\log\det(\mathbf{I}+\mathbf{H})}{\log(1+Tk)}$ and $S = \sqrt{\mathbf{y}^\top \mathbf{H}^{-1}\mathbf{y}}$.*

Theorem 5.1 provides a regret upper bound for PRB with the complexity of $\widetilde{\mathcal{O}}(\widetilde{d}\sqrt{kT})$ (see proofs in Appendix E). Instead, the graph-based methods (e.g., [18, 61]) lack an upper bound in terms of their performance. Theorem 5.1 provides insightful results in terms of PRB's performance. First, PRB's regret can grow sub-linearly with respective to $T$. Second, PRB's performance is affected by the number of nodes $k$. This indicates the larger the graph is, the more difficult the link prediction problem is. Third, $\widetilde{d}$ and $S$ in the regret upper bound reflect the complexity of the required neural function class to realize the underlying reward function $\mathbf{v}_t^*$, i.e., the difficulty of learning $\mathbf{v}_t^*$. $\widetilde{d}$ is the effective dimension, which measures the actual underlying dimension in the RKHS space spanned by NTK. $S$ is to provide an upper bound on the optimal parameters in the context of NTK regression. Both $\widetilde{d}$ and $S$ are two complexity terms that commonly exist in the literature of neural contextual bandits[76, 74]. In the general case when $1 > \alpha > 0$, learning $\mathbf{v}_t^*$ proportionally turns into a bandit optimization problem and the upper bound provided in Theorem 5.1 matches the SOTA results in neural bandits [76, 74]. In fact, the regret upper bound is closely related to the graph structure of $G_t$. In the special case when $\alpha = 1$, learning $\mathbf{v}_t^*$ turns into a simple convex optimization problem (Eq. (4.1)) and PRB can really find the optimal solution, which leads to zero regrets. When $\alpha = 0$, the problem turns into a complete bandit optimization problem with the same regret upper bound as Theorem 5.1.

## 6 Experiments

In this section, we begin by conducting a comprehensive evaluation of our proposed method, PRB, compared with both bandit-based and graph-based baselines across online and offline link prediction settings. Then, we analyze the computational costs associated with each experiment and present additional ablation studies related to PRB. In the implementation of PRB, we adapt the efficient PageRank algorithm [42] to solve Eq. (4.1).

### 6.1 Online Link Prediction

| Methods | MovieLens Mean ± Std | AmazonFashion Mean ± Std | Facebook Mean ± Std | GrQc Mean ± Std |
|---|---|---|---|---|
| EE-Net | 1638 ± 15.3 | 1698 ± 19.3 | 2274 ± 27.1 | 3419 ± 16.5 |
| NeuGreedy | 1955 ± 17.3 | 1952 ± 27.4 | 2601 ± 14.2 | 3629 ± 18.2 |
| NeuralUCB | 1737 ± 16.8 | 1913 ± 18.6 | 2190 ± 16.3 | 3719 ± 16.4 |
| NeuralTS | 1683 ± 14.7 | 2055 ± 21.9 | 2251 ± 19.5 | 3814 ± 23.3 |
| **PRB** | **1555 ± 21.7** | **1455 ± 18.4** | **1929 ± 17.0** | **3236 ± 18.5** |

Table 1: Cumulative regret of bandit-based methods on **online** link prediction.

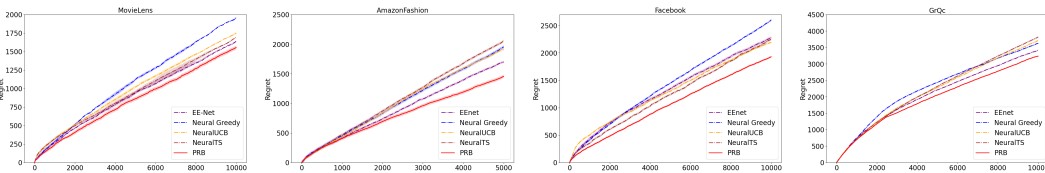

Figure 2: Regret comparison of bandit-based methods on **online** link prediction datasets (average of 10 runs with standard deviation in shadow, detailed in Table 1).

In this sub-section, we evaluate PRB on the setting of online link prediction and node classification as described in Sec. 3, compared with bandit-based baselines.

**Datasets and Setups.** We use three categories of real-world datasets to compare PRB with bandit-based baselines. The details and experiment settings are as follows.

(1) Recommendation datasets: Movielens [32] and Amazon Fashion [48] (Bipartite Graph). Given the user set $U$ and item set $I$, let $G_0$ be the graph with no edges, $G_0 = (V = U + I, E_0 = \emptyset)$. In round $t \in [T]$, we randomly select a user $v_t \in U$, and then randomly pick 100 items (arms) from $I$, including $v_t$'s 10 purchased items, forming $\mathcal{V}_t$. PRB runs based on $G_{t-1}$ and selects an arm (node) $v_{t,\hat{i}} \in \mathcal{V}_t$. If the selected arm $v_{t,\hat{i}}$ is the purchased item by $u_t$, the regret is 0 (or reward is 1) and we add the edge $[v_t, v_{t,\hat{i}}]$ to $G_{t-1}$, to form the new graph $G_t$; otherwise, the regret is 1 (or reward is 0) and $G_t = G_{t-1}$.

(2) Social network datasets: Facebook [38] and GR-QC [37]. Given the user set $V$, we have $G_0 = (V, E_0 = \emptyset)$. In a round $t \in [T]$, we randomly select a source node $v_t$ that can be thought of as the serving user. Then, we randomly choose 100 nodes, including $v_t$'s 10 connected nodes but their edges are removed, which form the arm pool $\mathcal{V}_t$ associated with the context set $\mathcal{X}_t$. Then, PRB will select one arm $v_{t,\hat{i}} \in \mathcal{V}_t$. If $v_t$ and $v_{t,\hat{i}}$ are connected in the original graph, the regret is 0 and add the edge $[v_t, v_{t,\hat{i}}]$ to $G_{t-1}$; otherwise, the regret is 1 and $G_t = G_{t-1}$.

(3) Node classification datasets: Cora, Citeseer, and Pubmed from the Planetoid citation networks [68]. Recall the problem setting described in Sec. 4. Consider a $k$-class node classification problem. Given a graph $G(V, E_0 = \emptyset)$, we randomly select a node $v_t \in V$ to predict its belonging class, in a round $t \in [T]$. Then, PRB select one super node $\tilde{v}_{i_t}$. If $v_t$ belongs to class $i_t$, the regret is 0 and add $[v_t, \tilde{v}_{i_t}]$ to $G_{t-1}$. Otherwise, the regret is 1 and $G_t = G_{t-1}$.

**Baselines.** For bandit-based methods, we apply Neural Greedy [12] that leverages the greedy exploration strategy on the exploitation network, NeuralUCB [76] that uses the exploitation network to learn the reward function along with an UCB-based exploration strategy, NeuralTS [74] that adopts the exploitation network to learn the reward function along with the Thompson Sampling exploration strategy, and EE-net [12] that utilizes the exploitation-exploration network to learn the reward function. Following [76, 12], for all methods, we train each network every 50 rounds for the first 2000 rounds and then every 100 rounds for the remaining rounds. See Appendix A.1 for additional experimental setups.

**Online Link Prediction**. We use Figure 2 to depict the regret trajectories over 10,000 rounds, and Table 1 to detail the cumulative regret after 10,000 rounds for all methods, where the lower is better. Based on the regret comparison, PRB consistently outperforms all other baselines across all datasets. For example, the cumulative regret at 10,000 rounds for PRB on MovieLens is considerably lower than the best-performing baseline, EE-Net. Similarly, in the AmazonFashion dataset, PRB achieved the lowest regret, surpassing the strongest baseline EE-Net over 14%. This trend is consistent across the Facebook and GrQc datasets, where PRB maintains its lead with the lowest regrets respectively. The consistency in PRB's performance across various datasets suggests the importance of utilizing the graph structure formed by previous link predictions.

| Methods | Cora | Citeseer | Pubmed |
| --- | --- | --- | --- |
| | Mean $\pm$ Std | Mean $\pm$ Std | Mean $\pm$ Std |
| EE-Net | $1990 \pm 13.8$ | $2299 \pm 33.4$ | $1659 \pm 11.3$ |
| NeuGreedy | $2826 \pm 21.4$ | $2543 \pm 24.6$ | $1693 \pm 13.5$ |
| NeuralUCB | $2713 \pm 21.7$ | $3101 \pm 22.0$ | $1672 \pm 14.3$ |
| NeuralTS | $1998 \pm 15.6$ | $3419 \pm 39.5$ | $1647 \pm 11.3$ |
| **PRB** | $\mathbf{1874 \pm 25.6}$ | $\mathbf{2168 \pm 35.7}$ | $\mathbf{1577 \pm 10.7}$ |

Table 2: Cumulative regret of bandit-based methods on **online** node classification.

**Online Node classification**. Figure 3 and Table 2 show the regret comparison on online node classification. PRB consistently demonstrates the lowest cumulative regret by outperforming other

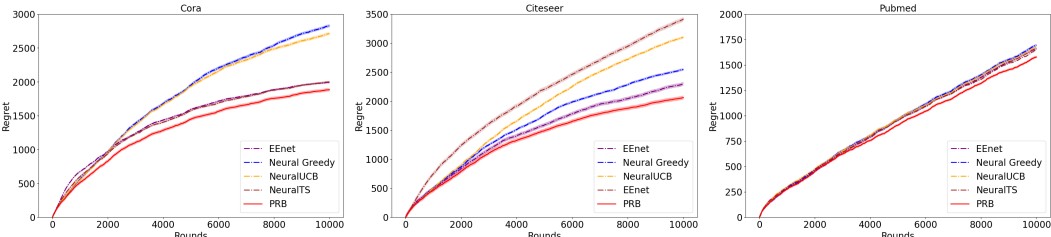

Figure 3: Regret comparison of bandit-based methods on **online** node classification datasets (average of 10 runs with standard deviation in shadow, detailed in Table 2.

| Methods | Cora
HR@100 ± Std | Citeseer
HR@100 ± Std | Pubmed
HR@100 ± Std | Collab
HR@50 ± Std | PPA
HR@100 ± Std | DDI
HR@20 ± Std |
|---|---|---|---|---|---|---|
| CN | 33.92 ± 0.46 | 29.79 ± 0.90 | 23.13 ± 0.15 | 56.44 ± 0.00 | 27.65 ± 0.00 | 17.73 ± 0.00 |
| AA | 39.85 ± 1.34 | 35.19 ± 1.33 | 27.38 ± 0.11 | 64.35 ± 0.00 | 32.45 ± 0.00 | 18.61 ± 0.00 |
| RA | 41.07 ± 0.48 | 33.56 ± 0.17 | 27.03 ± 0.35 | 64.00 ± 0.00 | 49.33 ± 0.00 | 27.60 ± 0.00 |
| GCN | 66.79 ± 1.65 | 67.08 ± 2.94 | 53.02 ± 1.39 | 44.75 ± 1.07 | 18.67 ± 1.32 | 37.07 ± 5.07 |
| SAGE | 55.02 ± 4.03 | 57.01 ± 3.74 | 39.66 ± 0.72 | 48.10 ± 0.81 | 16.55 ± 2.40 | 53.90 ± 4.74 |
| SEAL | 81.71 ± 1.30 | 83.89 ± 2.15 | 75.54 ± 1.32 | 64.74 ± 0.43 | 48.80 ± 3.16 | 30.56 ± 3.86 |
| NBFnet | 71.65 ± 2.27 | 74.07 ± 1.75 | 58.73 ± 1.99 | OOM | OOM | 4.00 ± 0.58 |
| Neo-GNN | 80.42 ± 1.31 | 84.67 ± 2.16 | 73.93 ± 1.19 | 57.52 ± 0.37 | 49.13 ± 0.60 | 63.57 ± 3.52 |
| BUDDY | 88.00 ± 0.44 | 92.93 ± 0.27 | 74.10 ± 0.78 | 65.94 ± 0.58 | 49.85 ± 0.20 | 78.51 ± 1.36 |
| NCN | 89.05 ± 0.96 | 91.56 ± 1.43 | 79.05 ± 1.16 | 64.76 ± 0.87 | 61.19 ± 0.85 | 82.32 ± 6.10 |
| NCNC | 89.65 ± 1.36 | 93.47 ± 0.95 | 81.29 ± 0.95 | 66.61 ± 0.71 | 61.42 ± 0.73 | 84.11 ± 3.67 |
| **PRB** | **92.33 ± 0.57** | **95.13 ± 1.28** | **84.54 ± 0.86** | **67.29 ± 0.31** | **63.47 ± 1.75** | **88.31 ± 4.36** |

Table 3: Results on **offline** link prediction benchmarks. OOM means out of GPU memory.

bandit methods at round 10,000, respectively. Overall, PRB decreases regrets by 3.0%, 1.2%, and 3.5% compared to one of the best baselines, NeuralTS. This experiment demonstrates that PRB is versatile enough for applications beyond online link prediction, extending to other real-world tasks such as online node classification. This highlights PRB's advantage of fusing contextual bandits with PageRank for collaborative exploitation and exploration.

## 6.2 Offline Link Prediction

In this subsection, we evaluate PRB in the setting of offline link prediction compared with graph-based baselines, where training and testing datasets are provided, following the same evaluation process of [18, 61]. Here, we train PRB on the training dataset using the same sequential optimization method Sec. 6.1. Then, we run the trained PRB on the testing dataset. Notice that PRB never sees the test data in the training process as other baselines.

**Datasets**. In this study, we use real-world link-prediction datasets to compare PRB with graph-based baselines. Specifically, we apply Cora, Citeseer, and Pubmed from Planetoid citation networks [68]; ogbl-collab, ogbl-ppa, and ogbl-ddi from Open Graph Benchmark [33]. (See dataset statistics in Appendix C.)

**Setting:** We strictly follow the experimental setup in [18] and use the Hits@k metric for evaluation. Please also refer to A.1 for additional setups.

**Baselines**. For graph-based methods, we choose traditional link-prediction heuristics including CN [15], RA [77], AA [3] and common GNNs including GCN [36] and SAGE [31]. Then, we employ SF-then-MPNN models, including SEAL [71] and NBFNet [78], as well as SF-and-MPNN models like Neo-GNN [70] and BUDDY[18]. Additionally, we also select the MPNN-then-SF model NCN [61] and NCNC [61]. The results of the baselines are sourced from Table 2 of [61].

**Comparison with Graph-based Baselines.** We present the experimental results in Table 3 for all methods. The results demonstrate that PRB consistently outperforms other baselines across all six datasets. Specifically, compared to the most recent method, NCNC, PRB achieves a minimum improvement of 0.68% on the Collab dataset, a maximum of 4.2%, and an average of 2.42% across

| Methods | MovieLens | AmazonFashion | Facebook | GrQc | Cora | Citeseer | Pubmed |
|---|---|---|---|---|---|---|---|
| | Mean $\pm$ Std | Mean $\pm$ Std | Mean $\pm$ Std | Mean $\pm$ Std | Mean $\pm$ Std | Mean $\pm$ Std | Mean $\pm$ Std |
| PRB | $1555 \pm 21.7$ | $1455 \pm 18.4$ | $1929 \pm 17.0$ | $3236 \pm 18.5$ | $1874 \pm 25.6$ | $2168 \pm 35.7$ | $\mathbf{1577 \pm 10.7}$ |
| PRB-Greedy | $1892 \pm 15.1$ | $1567 \pm 24.6$ | $1994 \pm 23.6$ | $3332 \pm 15.9$ | $1932 \pm 24.1$ | $2194 \pm 23.3$ | $1634 \pm 12.3$ |
| PRB-(10%-G) | $\mathbf{1521 \pm 17.6}$ | $\mathbf{1408 \pm 23.5}$ | $\mathbf{1858 \pm 15.7}$ | $\mathbf{3085 \pm 14.3}$ | $\mathbf{1804 \pm 23.5}$ | $\mathbf{2158 \pm 33.1}$ | $1630 \pm 11.5$ |

Table 4: Cumulative regrets of PRB variants for online link prediction and node classification.

all datasets. Given that all baselines lack the perspective of exploration, the results demonstrate that fusing the exploitation and exploration in contextual bandits along with learning graph connectivity through PageRank does significantly enhance accuracy for link prediction.

### 6.3 Ablation and Sensitivity Studies

Table 4 presents the performance of different variants of PRB, including PRB-greedy that only use the exploitation network and PRB-(10%-G) that has the warm start with addition 10% edges in $G_0$. The results show that exploration is crucial to the final performance and the additional graph knowledge can boost the performance.

Due to the space limit, we move all other experiment sections to Appendix A, including computational cost analysis on PRB and additional ablation & sensitivity studies.

## 7  Conclusion

This paper introduces a fusion algorithm for link prediction, which integrates the power of contextual bandits in balancing exploitation and exploration with propagation on graph structure by PageRank. We further provide the theoretical performance analysis for PRB, showing the regret of the proposed algorithm can grow sublinearly. We conduct extensive experiments in link prediction to evaluate PRB's effectiveness, compared with both bandit-based and graph-based baselines.

## Acknowledgments and Disclosure of Funding

This work is supported by the National Science Foundation under Award No. IIS-2117902 and DARPA (HR001121C0165). The views and conclusions are those of the authors and should not be interpreted as representing the official policies of the funding agencies or the government.

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

# A  Additional Experiments

## A.1  Experiment Setups

**Online Link Prediction Setups**. For all bandit-based methods including PRB, for fair comparison, the exploitation network $f_1$ is built by a 2-layer fully connected network with 100-width. For the exploration network of EE-Net and PRB, we use a 2-layer fully connected network with 100-width as well. For NeuralUCB and NeuralTS, following the setting of [76, 74], we use the exploitation network $f_1$ and conduct the grid search for the exploration parameter $\nu$ over $\{0.001, 0.01, 0.1, 1\}$ and for the regularization parameter $\lambda$ over $\{0.01, 0.1, 1\}$. For the neural bandits NeuralUCB/TS, following their setting, as they have expensive computation costs to store and compute the whole gradient matrix, we use a diagonal matrix to make an approximation. For all grid-searched parameters, we choose the best of them for comparison and report the average results of 10 runs for all methods. For all bandit-based methods, we use SGD as the optimizer for the exploitation network $f_1$. Additionally, for EE-Net and PRB, we use the Adam optimizer for the exploration network $f_2$. For all neural networks, we conduct the grid search for learning rate over $\{0.01, 0.001, 0.0005, 0.0001\}$. For PRB, we strictly follow the settings in [42] to implement the PageRank component. Specifically, we set the parameter $\alpha = 0.85$ after grid search over $\{0.1, 0.3, 0.5, 0.85, 0.9\}$, and the terminated accuracy $\epsilon = 10^{-6}$. For each dataset, we first shuffle the data and then run each network for 10,000 rounds ($t = 10,000$). We train each network every 50 rounds when $t < 2000$ and every 100 rounds when $2000 < t < 10,000$.

**Offline Link Prediction Setups**. For the graph-based methods, we strictly follow the experimental and hyperparameters settings in [61, 18] to reproduce the experimental results. Offline link prediction task requires graph links to play dual roles as both supervision labels and message passing links. For all datasets, the message-passing links at training time are equal to the supervision links, while at test and validation time, disjoint sets of links are held out for supervision that are never seen during training. All hyperparameters are tuned using Weights and Biases random search, exploring the search space of hidden dimensions from 64 to 512, dropout from 0 to 1, layers from 1 to 3, weight decay from 0 to 0.001, and learning rates from 0.0001 to 0.01. Hyperparameters yielding the highest validation accuracy are selected, and results are reported on a single-use test set. For PRB, we use setups similar to those in the online setting. We utilize the exploitation network $f_1$ and exploration network $f_2$ both with 500-width. We set the training epoch to 100 and evaluate the model performance on validation and test datasets. We utilize the Adam optimizer for all baseline models. For PRB implementation, We utilize the SGD optimizer for $f_1$ and the Adam optimizer for $f_2$.

## A.2  Computational Cost Analysis

We conduct all of our experiments on an Nvidia 3060 GPU with an x64-based processor.

**Time and space complexity**. Let $n$ be the number of nodes, $t$ be the index of the current round of link prediction, $k$ be the number of target candidate nodes, $d$ be the number of context dimensions, and $p$ be neural network width. For the online setting, let $m_t$ be the number of edges at round $t$. In the setting of online link prediction, the time complexity of PRB is $O(kdp + m_t k)$, where the first term is the cost of calculating the exploitation-exploration score for each candidate node and the second term is the cost of running PageRank, following [42]. And, the space complexity is $O(n + m_t)$ to store node weights and edges. For the offline setting, let $m$ be the number of edges in the testing dataset. Let $F$ be the number target links to predict. Then, the inference time complexity of PRB for $F$ links is $O(ndp) + \tilde{O}(mF)$. The first term is the cost of calculating the exploitation-exploration score for each node. The second term is the cost of PageRank [42]. The comparison with existing methods is listed in the following table:

In Figure 5, we analyze the running time of the internal components of PRB and PRB-Greedy (Algorithm 3). The comparison of the internal components reveals that the Random Walk phase accounts for 10% (PRB) and 6.3% (PRB-Greedy) on average of the total running time across seven datasets. Previous results also demonstrate that PRB significantly outperforms EE-Net which solely relies on the Exploitation-Exploration framework, by dedicating a small additional portion of time to the Random Walk component.

By recording the total training time of 10,000 rounds, we also compare PRB with other bandit-based baselines in Figure 4. Across all datasets, NeuralTS achieves the minimum average running time at 10.9 minutes, while PRB has the maximum at 17.5 minutes. Additionally, given that the Random

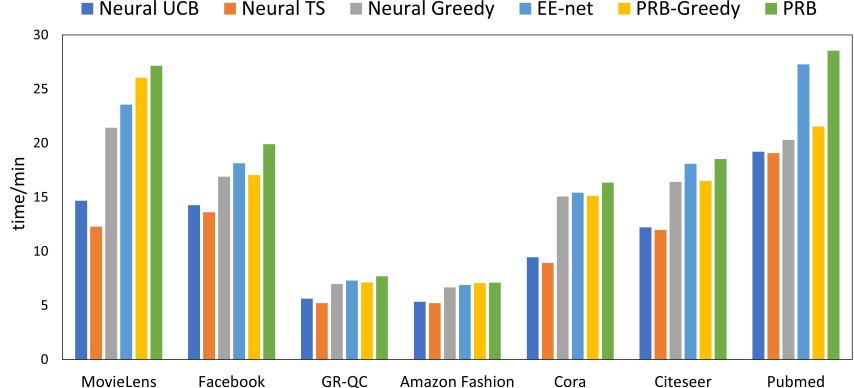

Figure 4: Running time comparison of PRB and bandit-based baselines.

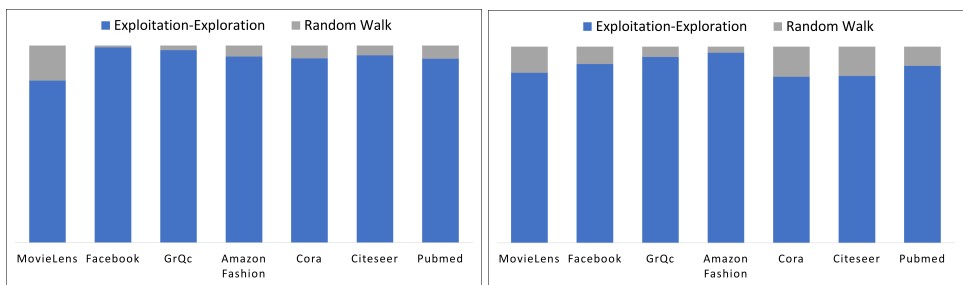

Figure 5: Proportion of running time for PRB-Greedy (left) and PRB (right) between exploitation-exploration and random walk.

Walk component takes only a minimal portion of our algorithm's running time, the average running times are relatively close between PRB-Greedy (15.4 minutes) and Neural Greedy (14.8 minutes), and between PRB (17.5 minutes) and EE-net (16.7 minutes). The comparative analysis reveals that while PRB incurs a relatively extended running time, it remains competitive with established baselines and demonstrates a significant enhancement in performance. This observation underscores the efficacy of PRB and supports its potential utility in practical applications despite its temporal demands.

Table 5 reports the inference time (one round in seconds) of bandit-based methods on three datasets for online link prediction. Although PRB takes a slightly longer time, it remains in the same order of magnitude as the other baselines. We adopt the approximated methods from [42] for the PageRank component to significantly reduce computation costs while ensuring good empirical performance.

Table 6 reports the inference time (one epoch of testing in seconds) of graph-based methods on three datasets for offline link prediction. PRB is faster than SEAL and shows competitive inference time as compared to other baselines.

| Methods | MovieLens | GrQc | Amazon |
|---|---|---|---|
| NeuralUCB | 0.11 | 0.01 | 0.02 |
| Neural Greedy | 0.14 | 0.02 | 0.03 |
| EE-Net | 0.17 | 0.03 | 0.04 |
| **PRB** | 0.20 | 0.03 | 0.04 |

Table 5: **Inference Time** (s) of PRB and bandit-based methods for online setting

| Methods | Cora | Pubmed | Collab |
|---|---|---|---|
| SEAL | 6.31 | 22.74 | 68.36 |
| Neo-GNN | 0.12 | 0.24 | 9.47 |
| BUDDY | 0.27 | 0.33 | 2.75 |
| NCNC | 0.04 | 0.07 | 1.58 |
| **PRB** | 0.11 | 0.58 | 3.52 |

Table 6: **Inference Time** (s) of PRB and graph-based methods for offline setting

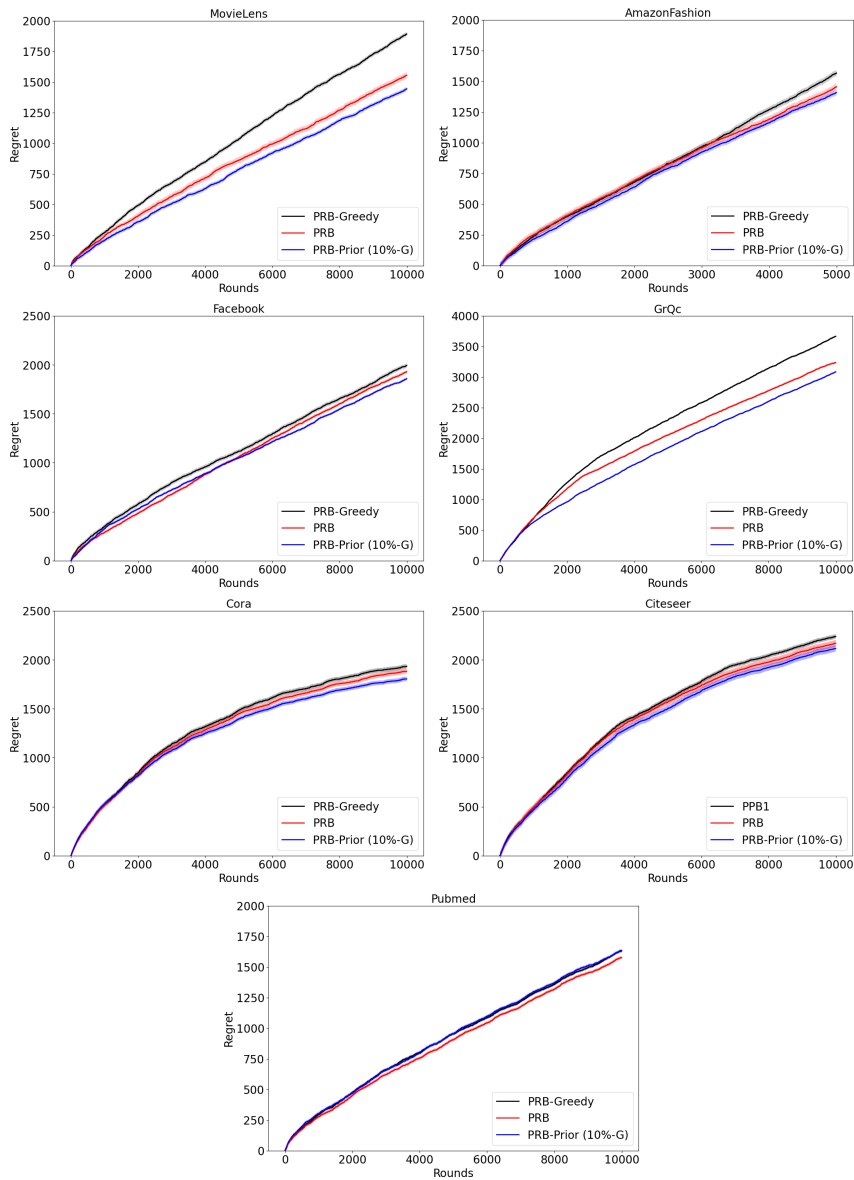

Figure 6: Regret Comparison of PRB-Greedy, PRB, and PRB-Prior (mean of 10 runs with standard deviation in shadow, detailed in Table 4 and 1).

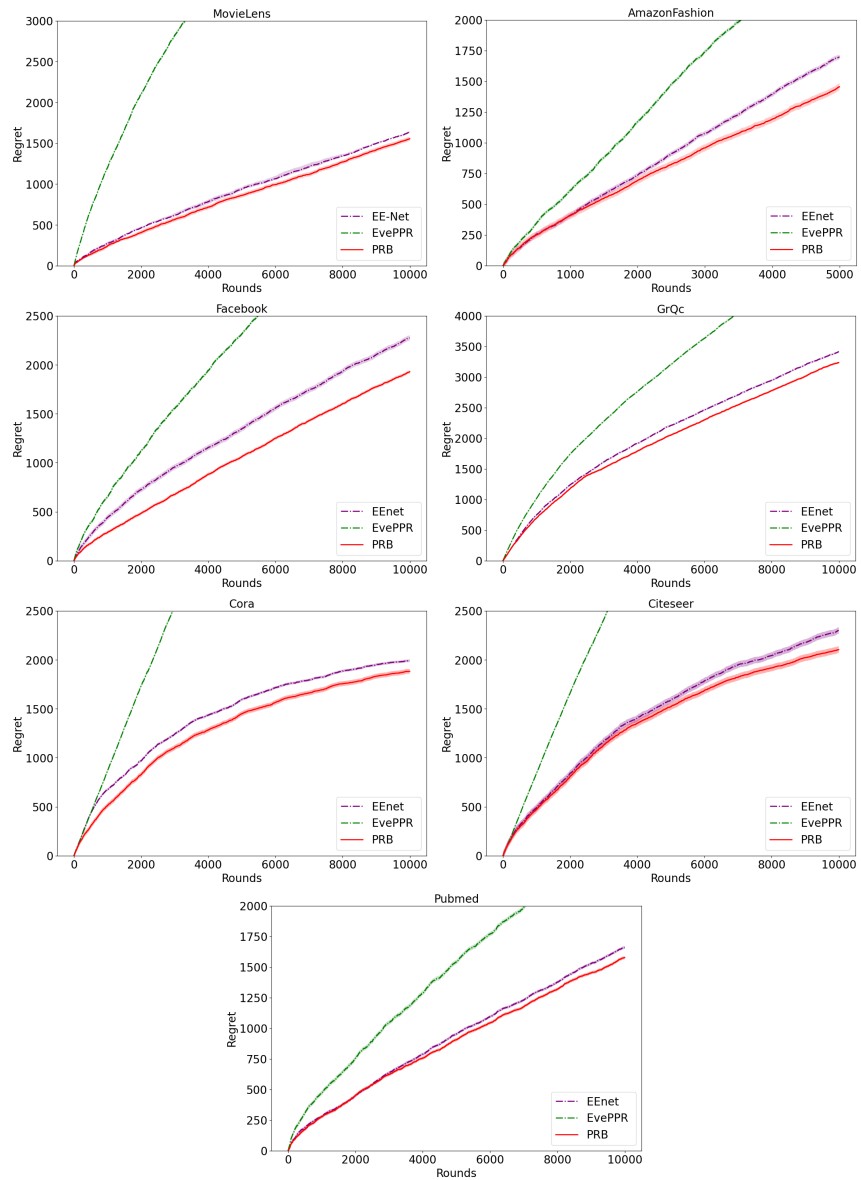

Figure 7: Regret Comparison of PRB, EEnet, and EvePPR (mean of 10 runs with standard deviation in shadow, detailed in Table 1).

### A.3 Additional Ablation and Sensitivity Studies

**PRB variants.** To extensively evaluate PRB in our experiments, we provide the following variants. PRB is the direct implementation of Algorithm 1. The initial graph $G_0$ only contains all nodes without any edges. PRB-Greedy is the greedy version of Algorithm 1 by removing the exploration network, as specified in Algorithm 3. PRB-Prior (10%-G) is Algorithm 1 with prior knowledge by revealing 10% of training edges on the initial graph. We apply PRB-Prior in our experiments to demonstrate how extra prior knowledge about the graph improves PRB's decision-making process.

Figure 6 and Table 4 highlights the regret comparison of three PRB variants: PRB, PRB-Greedy, and PRB-Prior. For both online link prediction and node classification, PRB surpasses PRB-Greedy by an average of 5.8%, highlighting the robustness of the exploration network embedded within PRB. Additionally, in online link prediction, the PRB-Prior (10%-G) variant consistently outperforms its counterparts across a majority of datasets. This is particularly evident in the MovieLens and AmazonFashion datasets, where it achieves notably low cumulative regrets of 1521 and 1408. Same in online node classification, PRB-Prior (10%-G) demonstrates exceptional performance on two out of three datasets, recording cumulative regrets of 1804 in Cora and 2158 in Citeseer. These results emphasize the benefits of incorporating prior knowledge within PRB to enhance predictive accuracy.

**Effectiveness of Bandits and PageRank.** In Figure 7, we compare the performance of PRB with that of EvePPR [42] and EE-Net [12], which represent methodologies based on PageRank and contextual bandits respectively. On one hand, PRB significantly outperforms EvePPR by integrating the exploitation and exploration strategy, which enhances PageRank's decision-making capabilities. On the other hand, PRB surpasses EE-net by leveraging a more comprehensive understanding of the input graph's structure and connectivity through enhanced PageRank. Overall, PRB consistently achieves lower regrets compared to both EvePPR and EE-Net, demonstrating the effectiveness of combining the exploitation-exploration with PageRank.

## B  Limitations

In this paper, we propose the PRB algorithm that integrates the exploitation-exploration of contextual bandits with PageRank. We do not investigate other integration methods, such as combining such exploitation-exploration with other Random Walk algorithms or GNNs. We also evaluate PRB on online link prediction and node classification. Several other real-world tasks, such as Subgraph Matching and Node Clustering, remain unexplored. Our future research will extend PRB to these and additional related tasks [79, 56] to assess its broader implications.

## C  Graph Dataset Statistics

|  | Cora | Citeseer | Pubmed | Collab | PPA | DDI |
|---|---|---|---|---|---|---|
| #Nodes | 2,708 | 3,327 | 18,717 | 235,868 | 576,289 | 4,267 |
| #Edges | 5,278 | 4,676 | 44,327 | 1,285,465 | 30,326,273 | 1,334,889 |
| Splits | random | random | random | fixed | fixed | fixed |
| Average Degree | 3.9 | 2.74 | 4.5 | 5.45 | 52.62 | 312.84 |

Table 7: Dataset Statistics

The statistics of each dataset are shown in Table 7. Random splits use 70%,10%, and 20% edges for training, validation, and test set respectively.

# D Variant Algorithms

---

**Algorithm 2** PRB-N (Node Classification)

---

**Input:** $f_1, f_2, T, G_0, \eta_1, \eta_2$ (learning rate)
1: Initialize $\theta_0^1, \theta_0^2$
2: **for** $t = 1, 2, \ldots, T$ **do**
3:      Observe serving node $v_t$, candidate nodes $\mathcal{V}_t = \{\tilde{v}_1, \tilde{v}_2, \ldots, \tilde{v}_k\}$, contexts $\mathcal{X}_t$ and Graph $G_{t-1}$
4:      **for** each $v_{t,i} \in \mathcal{V}_t$ **do**
5:          $\mathbf{h}_t[i] = f_1\left(x_{t,i}; \theta_{t-1}^1\right) + f_2\left(\phi\left(x_{t,i}\right); \theta_{t-1}^2\right)$
6:      **end for**
7:      Compute $\mathbf{P}_t$ based on $G_{t-1}$
8:      Solve $\mathbf{v}_t = \alpha \mathbf{P}_t \mathbf{v}_t + (1 - \alpha)\mathbf{h}_t$
9:      Select $\hat{i} = \arg\max_{v_{t,i} \in \mathcal{V}_t} \mathbf{v}_t[i]$
10:     Observe $r_{t,\hat{i}}$
11:     **if** $r_{t,\hat{i}} == 1$ **then**
12:         Add $[v_t, v_{t,\hat{i}}]$ to $G_{t-1}$ and set as $G_t$
13:     **else**
14:         $G_t = G_{t-1}$
15:     **end if**
16:     $\theta_t^1 = \theta_{t-1}^1 - \eta_1 \nabla_{\theta_{t-1}^1} \mathcal{L}\left(x_{t,\hat{i}}, r_{t,\hat{i}}; \theta_{t-1}^1\right)$
17:     $\theta_t^2 = \theta_{t-1}^2 - \eta_2 \nabla_{\theta_{t-1}^2} \mathcal{L}\left(\phi(x_{t,\hat{i}}), r_{t,\hat{i}} - f_1(x_{t,\hat{i}}; \theta_{t-1}^1); \theta_{t-1}^2\right)$
18: **end for**

---

---

**Algorithm 3** PRB-Greedy

---

**Input:** $f_1, f_2, T, G_0, \eta_1, \eta_2$ (learning rate)
1: Initialize $\theta_0^1, \theta_0^2$
2: **for** $t = 1, 2, \ldots, T$ **do**
3:      Observe serving node $v_t$, candidate nodes $\mathcal{V}_t$, contexts $\mathcal{X}_t$ and Graph $G_{t-1}$
4:      **for** each $v_{t,i} \in \mathcal{V}_t$ **do**
5:          $\mathbf{h}_t[i] = f_1\left(x_{t,i}; \theta_{t-1}^1\right)$
6:      **end for**
7:      Compute $\mathbf{P}_t$ based on $G_{t-1}$
8:      Solve $\mathbf{v}_t = \alpha \mathbf{P}_t \mathbf{v}_t + (1 - \alpha)\mathbf{h}_t$
9:      Select $\hat{i} = \arg\max_{v_{t,i} \in \mathcal{V}_t} \mathbf{v}_t[i]$
10:     Observe $r_{t,\hat{i}}$
11:     **if** $r_{t,\hat{i}} == 1$ **then**
12:         Add $[v_t, v_{t,\hat{i}}]$ to $G_{t-1}$ and set as $G_t$
13:     **else**
14:         $G_t = G_{t-1}$
15:     **end if**
16:     $\theta_t^1 = \theta_{t-1}^1 - \eta_1 \nabla_{\theta_{t-1}^1} \mathcal{L}\left(x_{t,\hat{i}}, r_{t,\hat{i}}; \theta_{t-1}^1\right)$
17: **end for**

---

# E  Proof of Theorem 5.1

## E.1  Preliminaries

Following neural function definitions and Lemmas of [13], given an instance $x$, we define the outputs of hidden layers of the neural network (Eq. (5.4)):

$$\mathbf{h}_0 = x, \mathbf{h}_l = \sigma(\mathbf{W}_l \mathbf{h}_{l-1}), l \in [L-1].$$

Then, we define the binary diagonal matrix functioning as ReLU:

$$\mathbf{D}_l = \text{diag}(\mathbb{1}\{(\mathbf{W}_l \mathbf{h}_{l-1})_1\}, \ldots, \mathbb{1}\{(\mathbf{W}_l \mathbf{h}_{l-1})_m\}), l \in [L-1].$$

Accordingly, the neural network (Eq. (5.4)) is represented by

$$f(x;\theta) = \mathbf{W}_L (\prod_{l=1}^{L-1} \mathbf{D}_l \mathbf{W}_l) x, \tag{E.1}$$

and

$$\nabla_{\mathbf{W}_l} f = \begin{cases} [\mathbf{h}_{l-1} \mathbf{W}_L (\prod_{\tau=l+1}^{L-1} \mathbf{D}_\tau \mathbf{W}_\tau)]^\top, l \in [L-1] \\ \mathbf{h}_{L-1}^\top, l = L. \end{cases} \tag{E.2}$$

Here, given a constant $R > 0$, we define the following function class:

$$B(\theta_0, R) = \{\theta \in \mathbb{R}^p : \|\theta - \theta_0\|_2 \le R/m^{1/4}\}. \tag{E.3}$$

Let $\mathcal{L}_t$ represent the squared loss function in round $t$. We use $x_1, x_2, \ldots, x_{Tk}$ represent all the context vectors presented in $T$ rounds. Then, we define the following instance-dependent complexity term:

$$\Psi(\theta_0, R) = \inf_{\theta \in \mathcal{B}(\theta_0, R)} \sum_{t=1}^{Tk} (f_2(x_t; \theta) - r_t)^2 \tag{E.4}$$

Then, we have the following auxiliary lemmas.

**Lemma E.1.** *Suppose $m, \eta_1, \eta_2$ satisfy the conditions in Theorem 5.1. With probability at least $1 - \mathcal{O}(TkL) \cdot \exp(-\Omega(m\omega^{2/3}L))$ over the random initialization, for all $t \in [T], i \in [k]$, $\theta$ satisfying $\|\theta - \theta_0\|_2 \le \omega$ with $\omega \le \mathcal{O}(L^{-9/2}[\log m]^{-3})$, it holds uniformly that*

$$(1), |f(x_{t,i}; \theta)| \le \mathcal{O}(1).$$
$$(2), \|\nabla_\theta f(x_{t,i}; \theta)\|_2 \le \mathcal{O}(\sqrt{L}).$$
$$(3), \|\nabla_\theta \mathcal{L}_t(\theta_t)\|_2 \le \mathcal{O}(\sqrt{L})$$

**Lemma E.2.** *Suppose $m, \eta_1, \eta_2$ satisfy the conditions in Theorem 5.1. With probability at least $1 - \mathcal{O}(TkL) \cdot \exp(-\Omega(m\omega^{2/3}L))$, for all $t \in [T], i \in [k]$, $\theta, \theta'$ (or $\Theta, \Theta'$) satisfying $\|\theta - \theta_0\|_2, \|\theta' - \theta_0\|_2 \le \omega$ with $\omega \le \mathcal{O}(L^{-9/2}[\log m]^{-3})$, it holds uniformly that*

$$|f(x; \theta) - f(x; \theta') - \langle \nabla_{\theta'} f(x; \theta'), \theta - \theta' \rangle| \le \mathcal{O}(w^{1/3} L^2 \sqrt{\log m}) \|\theta - \theta'\|_2.$$

**Lemma E.3.** *Suppose $m, \eta_1, \eta_2$ satisfy the conditions in Theorem 5.1. With probability at least $1 - \mathcal{O}(TkL) \cdot \exp(-\Omega(m\omega^{2/3}L))$, for all $t \in [T], i \in [k]$, $\theta, \theta'$ satisfying $\|\theta - \theta_0\|_2, \|\theta' - \theta_0\|_2 \le \omega$ with $\omega \le \mathcal{O}(L^{-9/2}[\log m]^{-3})$, it holds uniformly that*

$$(1) \qquad |f(x; \theta) - f(x; \theta')| \le \quad \mathcal{O}(\omega\sqrt{L}) + \mathcal{O}(\omega^{4/3} L^2 \sqrt{\log m}) \tag{E.5}$$

**Lemma E.4** (Almost Convexity)**.** *Let $\mathcal{L}_t(\theta) = (f(x_t; \theta) - r_t)^2/2$. Suppose $m, \eta_1, \eta_2$ satisfy the conditions in Theorem 5.1. With probability at least $1 - \mathcal{O}(TkL^2) \exp[-\Omega(m\omega^{2/3}L)]$ over randomness of $\theta_1$, for all $t \in [T]$, and $\theta, \theta'$ satisfying $\|\theta - \theta_0\|_2 \le \omega$ and $\|\theta' - \theta_0\|_2 \le \omega$ with $\omega \le \mathcal{O}(L^{-6}[\log m]^{-3/2})$, it holds uniformly that*

$$\mathcal{L}_t(\theta') \ge \mathcal{L}_t(\theta) + \langle \nabla_\theta \mathcal{L}_t(\theta), \theta' - \theta \rangle - \epsilon.$$

*where $\epsilon = \mathcal{O}(\omega^{4/3} L^3 \sqrt{\log m})$*

**Lemma E.5** (User Trajectory Ball). *Suppose $m, \eta_1, \eta_2$ satisfy the conditions in Theorem 5.1. With probability at least $1 - \mathcal{O}(TkL^2)\exp[-\Omega(m\omega^{2/3}L)]$ over randomness of $\theta_0$, for any $R > 0$, it holds uniformly that*

$$\|\theta_t - \theta_0\|_2 \leq \mathcal{O}(R/m^{1/4}), t \in [T].$$

**Lemma E.6** (Instance-dependent Loss Bound). *Let $\mathcal{L}_t(\theta) = (f(x_t; \theta) - r_t)^2/2$. Suppose $m, \eta_1, \eta_2$ satisfy the conditions in Theorem 5.1. With probability at least $1 - \mathcal{O}(TkL^2)\exp[-\Omega(m\omega^{2/3}L)]$ over randomness of $\theta_1$, given any $R > 0$ it holds that*

$$\sum_t^T \mathcal{L}_t(\theta_t) \leq \sum_t^T \mathcal{L}_t(\theta^*) + \mathcal{O}(1) + \frac{TLR^2}{\sqrt{m}} + \mathcal{O}(\frac{TR^{4/3}L^2\sqrt{\log m}}{m^{1/3}}). \tag{E.6}$$

*where $\theta^* = \arg\inf_{\theta \in B(\theta_0, R)} \sum_t^T \mathcal{L}_t(\theta)$.*

## E.2 Regret analysis

**Lemma E.7.** *Suppose $m, \eta_1, \eta_2$ satisfies the conditions in Theorem 5.1. In round $t \in [T]$, let $\hat{i}$ be the index selected by the algorithm. Then, For any $\delta \in (0, 1), R > 0$, with probability at least $1 - \delta$, for $t \in [T]$, it holds uniformly*

$$\frac{1}{t}\sum_{\tau=1}^t \mathbb{E}_{r_{\tau,\hat{i}}}\left[\left|f_1(x_{\tau,\hat{i}}; \theta^1_{\tau-1}) + f_2(\phi(x_{\tau,\hat{i}}); \theta^2_{\tau-1}) - r_{\tau,\hat{i}}\right| \mid \mathcal{H}_{\tau-1}\right]$$

$$\leq \frac{\sqrt{\Psi(\theta_0, R)} + \mathcal{O}(1)}{\sqrt{t}} + \sqrt{\frac{2\log(\mathcal{O}(1)/\delta)}{t}}. \tag{E.7}$$

*where $\mathcal{H}_t = \{x_{\tau,\hat{i}}, r_{\tau,\hat{i}}\}_{\tau=1}^t$ represents of historical data selected by $\pi_\tau$ and expectation is taken over the reward.*

*Proof.* First, according to Lemma E.5, $\theta_0^2, \ldots, \theta_{T-1}^2$ all are in $\mathcal{B}(\theta_0, R/m^{1/4})$. Then, according to Lemma E.1, for any $x \in \mathbb{R}^d, \|x\|_2 = 1$, it holds uniformly $|f_1(x_{t,\hat{i}}; \theta_t^1) + f_2(\phi(x_{t,\hat{i}}); \theta_t^2) - r_{t,\hat{i}}| \leq \mathcal{O}(1)$.

Then, for any $\tau \in [t]$, define

$$V_\tau := \mathbb{E}_{r_{\tau,\hat{i}}}\left[|f_2(\phi(x_{\tau,\hat{i}}); \theta^2_{\tau-1}) - (r_{\tau,\hat{i}} - f_1(x_{\tau,\hat{i}}; \theta^1_{\tau-1}))|\right]$$

$$- |f_2(\phi(x_{\tau,\hat{i}}); \theta^2_{\tau-1}) - (r_{\tau,\hat{i}} - f_1(x_{\tau,\hat{i}}; \theta^1_{\tau-1}))| \tag{E.8}$$

Then, we have

$$\mathbb{E}[V_\tau | F_{\tau-1}] = \mathbb{E}_{r_{\tau,\hat{i}}}\left[|f_2(\phi(x_{\tau,\hat{i}}); \theta^2_{\tau-1}) - (r_{\tau,\hat{i}} - f_1(x_{\tau,\hat{i}}; \theta^1_{\tau-1}))|\right]$$

$$- \mathbb{E}_{r_{\tau,\hat{i}}}\left[|f_2(\phi(x_{\tau,\hat{i}}); \theta^2_{\tau-1}) - (r_{\tau,\hat{i}} - f_1(x_{\tau,\hat{i}}; \theta^1_{\tau-1}))|\right] \tag{E.9}$$

$$= 0$$

where $F_{\tau-1}$ denotes the $\sigma$-algebra generated by the history $\mathcal{H}_{\tau-1}$.

Therefore, the sequence $\{V_\tau\}_{\tau=1}^t$ is the martingale difference sequence. Applying the Hoeffding-Azuma inequality, with probability at least $1 - \delta$, we have

$$\mathbb{P}\left[\frac{1}{t}\sum_{\tau=1}^t V_\tau - \underbrace{\frac{1}{t}\sum_{\tau=1}^t \mathbb{E}_{r_{i,\hat{i}}}[V_\tau | \mathbf{F}_{\tau-1}]}_{I_1} > \sqrt{\frac{2\log(1/\delta)}{t}}\right] \leq \delta \tag{E.10}$$

As $I_1$ is equal to 0, we have

$$\frac{1}{t}\sum_{\tau=1}^{t}\mathop{\mathbb{E}}_{r_{\tau,\hat{i}}}\left[\left|f_2(\phi(x_{\tau,\hat{i}});\theta^2_{\tau-1})-(r_{\tau,\hat{i}}-f_1(x_{\tau,\hat{i}};\theta^1_{\tau-1}))\right|\right]$$

$$\leq\underbrace{\frac{1}{t}\sum_{\tau=1}^{t}\left|f_2(\phi(x_{\tau,\hat{i}});\theta^2_{\tau-1})-(r_{\tau,\hat{i}}-f_1(x_{\tau,\hat{i}};\theta^1_{\tau-1}))\right|}_{I_3}+\sqrt{\frac{2\log(1/\delta)}{t}}\ . \tag{E.11}$$

For $I_3$, based on Lemma E.6, for any $\theta'$ satisfying $\|\theta'-\theta^2_0\|_2\leq R/m^{1/4}$, with probability at least $1-\delta$, we have

$$I_3\leq\frac{1}{t}\sqrt{t}\sqrt{\sum_{\tau=1}^{t}\left(f_2(\phi(x_{\tau,\hat{i}});\theta^2_{\tau-1})-(r_{\tau,\hat{i}}-f_1(x_{\tau,\hat{i}};\theta^1_{\tau-1}))\right)^2}$$

$$\leq\frac{1}{t}\sqrt{t}\sqrt{\sum_{\tau=1}^{t}\left(f_2(\phi(x_{\tau,\hat{i}});\theta')-(r_{\tau,\hat{i}}-f_1(x_{\tau,\hat{i}};\theta^1_{\tau-1}))\right)^2}+\frac{\mathcal{O}(1)}{\sqrt{t}} \tag{E.12}$$

$$\overset{(a)}{\leq}\frac{\sqrt{\Psi(\theta_0,R)}+\mathcal{O}(1)}{\sqrt{t}}.$$

where $(a)$ is based on the definition of instance-dependent complexity term. Combining the above inequalities together, with probability at least $1-\delta$, we have

$$\frac{1}{t}\sum_{\tau=1}^{t}\mathop{\mathbb{E}}_{r_{\tau,\hat{i}}}\left[\left|f_2(\phi(x_{\tau,\hat{i}});\theta^2_{\tau-1})-(r_{\tau,\hat{i}}-f_1(x_{\tau,\hat{i}};\theta^1_{\tau-1})\right|\right]$$

$$\leq\frac{\sqrt{\Psi(\theta_0,R)}+\mathcal{O}(1)}{\sqrt{t}}+\sqrt{\frac{2\log(\mathcal{O}(1)/\delta)}{t}}. \tag{E.13}$$

The proof is completed. $\qquad\square$

**Corollary E.1.** *Suppose $m,\eta_1,\eta_2$ satisfy the conditions in Theorem 5.1. For any $t\in[T]$, let $i^*$ be the index selected by some fixed policy and $r_{t,i^*}$ is the corresponding reward, and denote the policy by $\pi^*$. Let $\theta^{1,*}_{t-1},\theta^{2,*}_{t-1}$ be the intermediate parameters trained by Algorithm 1 using the data select by $\pi^*$. Then, with probability at least $(1-\delta)$ over the random of the initialization, for any $\delta\in(0,1),R>0$, it holds that*

$$\frac{1}{t}\sum_{\tau=1}^{t}\mathop{\mathbb{E}}_{r_{\tau,i^*}}\left[\left|f_2(\phi(x_{\tau,i^*});\theta^{2,*}_{\tau-1})-\left(r_{\tau,i^*}-f_1(x_{\tau,i^*};\theta^{1,*}_{\tau-1})\right)\right|\mid\pi^*,\mathcal{H}^*_{\tau-1}\right]$$

$$\leq\frac{\sqrt{\Psi(\theta_0,R)}+\mathcal{O}(1)}{\sqrt{t}}+\sqrt{\frac{2\log(\mathcal{O}(1)/\delta)}{t}}, \tag{E.14}$$

*where $\mathcal{H}^*_{\tau-1}=\{x_{\tau,i^*},r_{\tau,i^*}\}_{\tau'=1}^{\tau-1}$ represents the historical data produced by $\pi^*$ and the expectation is taken over the reward.*

Define $g(x_t;\theta)\nabla_\theta=f(x_t;\theta)$ for brevity.

**Lemma E.8.** *Suppose $m$ satisfies the conditions in Theorem 5.1. With probability at least $1-\delta$ over the initialization, there exists $\theta'\in B(\theta_0,\widetilde{\Omega}(T^{3/2}))$, such that*

$$\sum_{t=1}^{Tk}\mathbb{E}[(r_t-f(x_t;\theta'))^2/2]\leq\mathcal{O}\left(\sqrt{\widetilde{d}\log(1+Tk)-2\log\delta}+S+1\right)^2\cdot\widetilde{d}\log(1+Tk).$$

*Proof.*

$$\mathbb{E}[\sum_{t=1}^{Tk}(r_t - f(x_t; \theta'))^2]$$

$$= \sum_{t=1}^{TK}(y(x_t) - f(x_t; \theta'))^2$$

$$\overset{(a)}{\leq} \mathcal{O}\left(\sqrt{\log\left(\frac{\det(\mathbf{A}_T)}{\det(\mathbf{I})}\right) - 2\log\delta + S + 1}\right)^2 \sum_{t=1}^{TK} \|g(x_t; \theta_0)\|_{\mathbf{A}_T^{-1}}^2 + 2TK \cdot \mathcal{O}\left(\frac{T^2 L^3 \sqrt{\log m}}{m^{1/3}}\right)$$

$$\overset{(b)}{\leq} \mathcal{O}\left(\sqrt{\widetilde{d}\log(1 + Tk) - 2\log\delta + S + 1}\right)^2 \cdot \left(\widetilde{d}\log(1 + Tk) + 1\right) + \mathcal{O}(1),$$

where $(a)$ is based on Lemma E.9 and $(b)$ is an application of Lemma 11 in [1] and Lemma E.12, and $\mathcal{O}(1)$ is induced by the choice of $m$. By ignoring $\mathcal{O}(1)$, The proof is completed. $\square$

**Definition E.1.** *Given the context vectors $\{x_i\}_{i=1}^T$ and the rewards $\{r_i\}_{i=1}^T$, then we define the estimation $\widehat{\theta}_t$ via ridge regression:*

$$\mathbf{A}_t = \mathbf{I} + \sum_{i=1}^t g(x_i; \theta_0)g(x_i; \theta_0)^\top$$

$$\mathbf{b}_t = \sum_{i=1}^t r_i g(x_i; \theta_0)$$

$$\widehat{\theta}_t = \mathbf{A}_t^{-1}\mathbf{b}_t$$

**Lemma E.9.** *Suppose $m$ satisfies the conditions in Theorem 5.1. With probability at least $1 - \delta$ over the initialization, there exists $\theta' \in B(\theta_0, \widetilde{\Omega}(T^{3/2}))$ for all $t \in [T]$, such that*

$$|y(x_t) - f(x_t; \theta')| \leq \mathcal{O}\left(\sqrt{\log\left(\frac{\det(\mathbf{A}_t)}{\det(\mathbf{I})}\right) - 2\log\delta + S + 1}\right)\|g(x_t; \theta_0)\|_{\mathbf{A}_t^{-1}} + \mathcal{O}\left(\frac{T^2 L^3 \sqrt{\log m}}{m^{1/3}}\right)$$

*Proof.* Given a set of context vectors $\{x\}_{t=1}^T$ with the ground-truth function $h$ and a fully-connected neural network $f$, we have

$$|y(x_t) - f(x_t; \theta')|$$

$$\leq \left|y(x_t) - \langle g(x_t; \theta_0), \widehat{\theta}_t\rangle\right| + \left|f(x_t; \theta') - \langle g(x_t; \theta_0), \widehat{\theta}_t\rangle\right|$$

where $\theta'$ is the estimation of ridge regression from Definition E.1. Then, based on the Lemma E.10, there exists $\theta^* \in \mathbf{R}^P$ such that $h(x_i) = \langle g(x_i, \theta_0), \theta^*\rangle$. Thus, we have

$$\left|y(x_t) - \langle g(x_t; \theta_0), \widehat{\theta}_t\rangle\right|$$

$$= \left|\langle g(x_i, \theta_0), \theta^*\rangle - \left\langle g(x_i, \theta_0), \widehat{\theta}_t\right\rangle\right|$$

$$\leq \mathcal{O}\left(\sqrt{\log\left(\frac{\det(\mathbf{A}_t)}{\det(\mathbf{I})}\right) - 2\log\delta + S}\right)\|g(x_t; \theta_0)\|_{\mathbf{A}_t^{-1}}$$

where the final inequality is based on the the Theorem 2 in [1], with probability at least $1 - \delta$, for any $t \in [T]$.

Second, we need to bound

$$\left|f(x_t; \theta') - \langle g(x_t; \theta_0), \widehat{\theta}_t\rangle\right|$$

$$\leq |f(x_t; \theta') - \langle g(x_t; \theta_0), \theta' - \theta_0\rangle|$$

$$\quad + \left|\langle g(x_t; \theta_0), \theta' - \theta_0\rangle - \langle g(x_t; \theta_0), \widehat{\theta}_t\rangle\right|$$

To bound the above inequality, we first bound

$$|f(x_t; \theta') - \langle g(x_t; \theta_0), \theta' - \theta_0 \rangle|$$
$$= |f(x_t; \theta') - f(\mathbf{x}_t; \theta_0) - \langle g(x_t; \theta_0), \theta' - \theta_0 \rangle|$$
$$\leq \mathcal{O}(\omega^{4/3} L^3 \sqrt{\log m})$$

where we initialize $f(\mathbf{x}_t; \theta_0) = 0$ and the inequality is derived by Lemma E.2 with $\omega = \frac{\mathcal{O}(t^{3/2})}{m^{1/4}}$.
Next, we need to bound

$$|\langle g(x_t; \theta_0), \theta' - \theta_0 \rangle - \langle g(x_t; \theta_0), \widehat{\theta}_t \rangle|$$
$$= |\langle g(x_t; \theta_0), (\theta' - \theta_0 - \widehat{\theta}_t) \rangle|$$
$$\leq \|g(x_t; \theta_0)\|_{\mathbf{A}_t^{-1}} \cdot \|\theta' - \theta_0 - \widehat{\theta}_t\|_{\mathbf{A}_t}$$
$$\leq \|g(x_t; \theta_0)\|_{\mathbf{A}_t^{-1}} \cdot \|\mathbf{A}_t\|_2 \cdot \|\theta' - \theta_0 - \widehat{\theta}_t\|_2.$$

Due to the Lemma E.12 and Lemma E.11, we have

$$\|\mathbf{A}_t\|_2 \cdot \|\theta' - \theta_0 - \widehat{\theta}_t\|_2 \leq (1 + t\mathcal{O}(L)) \cdot \frac{1}{1 + \mathcal{O}(tL)} = \mathcal{O}(1).$$

Finally, putting everything together, we have

$$|y(x_t) - f(x_t; \theta')| \leq \gamma_1 \|g(x_t; \theta_0)/\sqrt{m}\|_{\mathbf{A}_t^{-1}} + \gamma_2.$$

The proof is completed. □

**Definition E.2.**
$$\mathbf{G}^{(0)} = [g(x_1; \theta_0), \ldots, g(x_T; \theta_0)] \in \mathbb{R}^{p \times T}$$
$$\mathbf{G}_0 = [g(x_1; \theta_0), \ldots, g(x_{Tk}; \theta_0)] \in \mathbb{R}^{p \times Tk}$$
$$\mathbf{r} = (r_1, \cdots, r_T) \in \mathbb{R}^T$$

$\mathbf{G}^{(0)}$ *and* $\mathbf{r}$ *are formed by the selected contexts and observed rewards in* $T$ *rounds,* $\mathbf{G}_0$ *are formed by all the presented contexts.*

*Inspired by Lemma B.2 in [76] , with* $\eta = m^{-1/4}$ *we define the auxiliary sequence following :*

$$\theta_0 = \theta^{(0)}, \quad \theta^{(j+1)} = \theta^{(j)} - \eta \left[ \mathbf{G}^{(0)} \left( [\mathbf{G}^{(0)}]^\top (\theta^{(j)} - \theta_0) - \mathbf{r} \right) + \lambda(\theta^{(j)} - \theta_0) \right]$$

**Lemma E.10.** *Suppose* $m$ *satisfies the conditions in Theorem 5.1. With probability at least* $1 - \delta$ *over the initialization, for any* $t \in [T], i \in [k]$*, the result uniformly holds:*

$$h_{u_t}(x_{t,i}) = \langle g(x_{t,i}; \theta_0), \theta^* - \theta_0 \rangle.$$

*Proof.* Based on Lemma E.13 with proper choice of $\epsilon$, we have

$$\mathbf{G}_0^\top \mathbf{G}_0 \succeq \mathbf{H} - \|\mathbf{G}_0^\top \mathbf{G}_0 - \mathbf{H}\|_F \mathbf{I} \succeq \mathbf{H} - \lambda_0 \mathbf{I}/2 \succeq \mathbf{H}/2 \succeq 0.$$

Define $\mathbf{h} = [h_{u_1}(x_1), \ldots, h_{u_T}(x_{Tk})]$. Suppose the singular value decomposition of $\mathbf{G}_0$ is $\mathbf{PAQ}^\top, \mathbf{P} \in \mathbb{R}^{p \times Tk}, \mathbf{A} \in \mathbb{R}^{Tk \times Tk}, \mathbf{Q} \in \mathbb{R}^{Tk \times Tk}$, then, $\mathbf{A} \succeq 0$. Define $\theta^* = \theta_0 + \mathbf{PA}^{-1}\mathbf{Q}^\top \mathbf{h}$.
Then, we have
$$\mathbf{G}_0^\top (\theta^* - \theta_0) = \mathbf{QAP}^\top \mathbf{PA}^{-1} \mathbf{Q}^\top \mathbf{h} = \mathbf{h}.$$

which leads to

$$\sum_{t=1}^{T} \sum_{i=1}^{k} (h_{u_t}(x_{t,i}) - \langle g(x_{t,i}; \theta_0), \theta^* - \theta_0 \rangle) = 0.$$

Therefore, the result holds:

$$\|\theta^* - \theta_0\|_2^2 = \mathbf{h}^\top \mathbf{QA}^{-2} \mathbf{Q}^\top \mathbf{h} = \mathbf{h}^\top (\mathbf{G}_0^\top \mathbf{G}_0)^{-1} \mathbf{h} \leq 2\mathbf{h}^\top \mathbf{H}^{-1} \mathbf{h} \tag{E.15}$$

□

**Lemma E.11.** *There exist $\theta' \in B(\theta_0, \widetilde{\mathcal{O}}(T^{3/2}L + \sqrt{T}))$, such that, with probability at least $1 - \delta$, the results hold:*

$$(1) \|\theta' - \theta_0\|_2 \leq \frac{\widetilde{\mathcal{O}}(T^{3/2}L + \sqrt{T})}{m^{1/4}}$$

$$(2) \|\theta' - \theta_0 - \widehat{\theta}_t\|_2 \leq \frac{1}{1 + \mathcal{O}(TL)}$$

*Proof.* The sequence of $\theta^{(j)}$ is updates by using gradient descent on the loss function:

$$\min_\theta \mathcal{L}(\theta) = \frac{1}{2}\|[\mathbf{G}^{(0)}]^\top(\theta - \theta^{(0)}) - \mathbf{r}\|_2^2 + \frac{m\lambda}{2}\|\theta - \theta^{(0)}\|_2^2.$$

For any $j > 0$, the results holds:

$$\|\mathbf{G}^{(0)}\|_F \leq \sqrt{T}\max_{t \in [T]}\|g(x_t; \theta_0)\|_2 \leq \mathcal{O}(\sqrt{TL}),$$

where the last inequality is held by Lemma E.1. Finally, given the $j > 0$,

$$\|\theta^{(j)} - \theta^{(0)}\|_2^2 \leq \sum_{i=1}^j \eta\left[\mathbf{G}^{(0)}\left([\mathbf{G}^{(0)}]^\top(\theta^{(i)} - \theta_0) - \mathbf{r}\right) + \lambda(\theta^{(i)} - \theta_0)\right] \leq \frac{\mathcal{O}(j(TL\sqrt{T/\lambda} + \sqrt{T\lambda}))}{m^{1/4}}.$$

$$\text{(E.16)}$$

For (2), by standard results of gradient descent on ridge regression, $\theta^{(j)}$, and the optimum is $\theta^{(0)} + \widehat{\theta}_t$. Therefore, we have

$$\|\theta^{(j)} - \theta^{(0)} - \widehat{\theta}_t\|_2^2 \leq [1 - \eta\lambda]^j \frac{2}{\lambda}\left(\mathcal{L}(\theta^{(0)}) - \mathcal{L}(\theta^{(0)} + \widehat{\theta}_t)\right)$$

$$\leq \frac{2(1 - \eta\lambda)^j}{\lambda}\mathcal{L}(\theta^{(0)})$$

$$= \frac{2(1 - \eta m\lambda)^j}{\lambda}\frac{\|\mathbf{r}\|_2^2}{2}$$

$$\leq \frac{T(1 - \eta\lambda)^j}{\lambda}.$$

By setting $\lambda = 1$ and $j = \log((T + \mathcal{O}(T^2L))^{-1})/\log(1 - m^{-1/4})$, we have $\|\theta^{(j)} - \theta_0 - \widehat{\theta}_t\|_2^2 \leq \frac{1}{1+\mathcal{O}(TL)}$. Replacing $k$ and $\lambda$ in (E.16) finishes the proof. $\square$

**Lemma E.12.** *Suppose $m$ satisfies the conditions in Theorem 5.1. With probability at least $1 - \delta$ over the initialization, the result holds:*

$$\|\mathbf{A}_T\|_2 \leq 1 + \mathcal{O}(TL),$$

$$\log\frac{\det \mathbf{A}_T}{\det \mathbf{I}} \leq \widetilde{d}\log(1 + Tk) + 1.$$

*Proof.* Based on the Lemma E.1, for any $t \in [T]$, $\|g(x_t; \theta_0)\|_2 \leq \mathcal{O}(\sqrt{L})$. Then, for the first item:

$$\|\mathbf{A}_T\|_2 = \|\mathbf{I} + \sum_{t=1}^T g(x_t; \theta_0)g(x_t; \theta_0)^\top\|_2$$

$$\leq \|\mathbf{I}\|_2 + \|\sum_{t=1}^T g(x_t; \theta_0)g(x_t; \theta_0)^\top\|_2$$

$$\leq 1 + \sum_{t=1}^T \|g(x_t; \theta_0)\|_2^2 \leq 1 + \mathcal{O}(TL).$$

Next, we have

$$\log\frac{\det(\mathbf{A}_T)}{\det(\mathbf{I})} = \log\det(\mathbf{I} + \sum_{t=1}^{Tk} g(x_t; \theta_0)g(x_t; \theta_0)^\top) = \det(\mathbf{I} + \mathbf{G}_0\mathbf{G}_0^\top)$$

Then, we have

$$\log\det(\mathbf{I} + \mathbf{G}_0\mathbf{G}_0^\top)$$
$$= \log\det(\mathbf{I} + \mathbf{H} + (\mathbf{G}_0\mathbf{G}_0^\top - \mathbf{H}))$$
$$\leq \log\det(\mathbf{I} + \mathbf{H}) + \langle(\mathbf{I} + \mathbf{H})^{-1}, (\mathbf{G}_0\mathbf{G}_0^\top - \mathbf{H})\rangle$$
$$\leq \log\det(\mathbf{I} + \mathbf{H}) + \|(\mathbf{I} + \mathbf{H})^{-1}\|_F\|\mathbf{G}_0\mathbf{G}_0^\top - \mathbf{H}\|_F$$
$$\leq \log\det(\mathbf{I} + \mathbf{H}) + \sqrt{T}\|\mathbf{G}_0\mathbf{G}_0^\top - \mathbf{H}\|_F$$
$$\leq \log\det(\mathbf{I} + \mathbf{H}) + 1$$
$$= \widetilde{d}\log(1 + Tk) + 1.$$

The first inequality is because the concavity of $\log\det$ ; The third inequality is due to $\|(\mathbf{I} + \mathbf{H}\lambda)^{-1}\|_F \leq \|\mathbf{I}^{-1}\|_F \leq \sqrt{T}$; The last inequality is because of the choice the $m$, based on Lemma E.13; The last equality is because of the Definition of $\widetilde{d}$. The proof is completed. $\square$

**Lemma E.13.** *For any $\delta \in (0, 1)$, if $m = \Omega\left(\frac{L^6\log(TkL/\delta)}{(\epsilon/Tk)^4}\right)$, then with probability at least $1 - \delta$, the results hold:*

$$\|\mathbf{G}_0\mathbf{G}_0^\top - \mathbf{H}\|_F \leq \epsilon.$$

*Proof.* This is an application of Lemma B.1 in [76] by properly setting $\epsilon$. $\square$

**Lemma E.14** (Exactness of PageRank [42])**.** *When PageRank achieves the stationary distribution,* $\mathbf{v}_t = \frac{1-\alpha}{\mathbf{I}-\alpha\mathbf{P}_t}\mathbf{h}_t$.

Finally, we provide the proof of Theorem 5.1.

*Proof.*

$$\mathbf{v}_t^*[i^*] - \mathbf{v}_t^*[\hat{i}]$$
$$= \mathbf{v}_t^*[i^*] - \mathbf{v}_t[\hat{i}] + \mathbf{v}_t[\hat{i}] - \mathbf{v}_t^*[\hat{i}]$$
$$\overset{(1)}{\leq} \mathbf{v}_t^*[i^*] - \mathbf{v}_t[i^*] + \mathbf{v}_t[\hat{i}] - \mathbf{v}_t^*[\hat{i}]$$
$$\leq |\mathbf{v}_t^*[i^*] - \mathbf{v}_t[i^*]| + |\mathbf{v}_t[\hat{i}] - \mathbf{v}_t^*[\hat{i}]|$$
$$\leq 2\|\mathbf{v}_t^* - \mathbf{v}_t\|_2$$
$$\overset{(2)}{=} 2\left\|\frac{1-\alpha}{\mathbf{I}-\alpha\mathbf{P}_t}\mathbf{y}_t - \frac{1-\alpha}{\mathbf{I}-\alpha\mathbf{P}_t}\mathbf{h}_t\right\|_2$$
$$\leq 2\left\|\frac{1-\alpha}{\mathbf{I}-\alpha\mathbf{P}_t}\right\|_2\|\mathbf{y}_t - \mathbf{h}_t\|_2$$

where (1) is by the choice of PRB and (2) is based on the exact solution of PageRank [42]. Let $\lambda_{\max}$ be the maximal eigenvalue of $\mathbf{P}_t$. Because $\mathbf{P}_t$ is a stochastic matrix, $\lambda_{\max} = 1$. Then, we have

$$\mathbf{I} - \alpha\mathbf{P}_t \succeq (1 - \alpha\lambda_{\max})\mathbf{I} \succeq (1 - \alpha)\mathbf{I}.$$

Accordingly, we have

$$\left\|\frac{1-\alpha}{\mathbf{I}-\alpha\mathbf{P}_t}\right\|_2\|\mathbf{y}_t - \mathbf{h}_t\|_2 \leq \left\|\frac{1-\alpha}{1-\alpha}\mathbf{I}\right\|_2\|\mathbf{y}_t - \mathbf{h}_t\|_2 = \|\mathbf{y}_t - \mathbf{h}_t\|_2.$$

Then, based on Corollary E.1, Lemma E.8, and Lemma E.3, with shadow parameters, we have

$$
\|\mathbf{y}_t - \mathbf{h}_t\|_2
$$

$$
= \sqrt{\sum_{v_{t,i} \in \mathcal{V}_t} [y(x_{t,i}) - (f_2(\phi(x_{t,i}); \theta_{t-1}^2) + f_1(x_{t,i}; \theta_{t-1}^1))]^2}
$$

$$
= \sqrt{\sum_{v_{t,i} \in \mathcal{V}_t} [f_2(\phi(x_{t,i}); \theta_{t-1}^2) - (y(x_{t,i}) - f_1(x_{t,i}; \theta_{t-1}^1))]^2}
$$

$$
= \sqrt{k \left[ \frac{\widetilde{\mathcal{O}}(\sqrt{\Psi(\theta_0, R)})}{\sqrt{t}} \right]^2}
$$

$$
\leq \widetilde{\mathcal{O}}(\sqrt{\tilde{d}k/T}) \cdot \sqrt{\max(\tilde{d}, S^2)}.
$$

To sum up, we have

$$
\mathbf{R}_T = \sum_{t=1}^{T} (\mathbf{v}_t^*[i^*] - \mathbf{v}_t^*[\hat{i}])
$$

$$
\leq \sum_{t=1}^{T} \|\mathbf{y}_t - \mathbf{h}_t\|_2
$$

$$
\leq \widetilde{\mathcal{O}}(\sqrt{\tilde{d}kT}) \cdot \sqrt{\max(\tilde{d}, S^2)}
$$

The proof is completed. $\qquad\square$

