# OpenReview forum: "PageRank Bandits for Link Prediction"
_NeurIPS.cc/2024/Conference — NeurIPS 2024 poster_

### Official Review · Reviewer_bPwg · 2024-06-20

**Soundness:** 2
**Presentation:** 3
**Contribution:** 2
**Rating:** 5
**Confidence:** 4

**Summary:**

The authors propose PRB, a new method that blends exploration, exploitation from previous neural bandit literature into an architecture that effectively considers graph connectivity in order to boost the performance for both the node classification and link prediction tasks. The authors demonstrate the soundness of the new model with proofs and mathematical reasoning, which also distinguishes it from previous neural bandit research. Additionally, the authors verify the effectiveness of the model by measuring it's performance in online and offline link prediction along with node classification.

**Strengths:**

* Well-written paper that explains PRB's functionality and also highlights the difference between the problem solved by this paper and other neural bandit papers.
* PRB empirically tested against SOTA baselines across online and offline link prediction and tested on node classification.

**Weaknesses:**

* Clarity of descriptions is sometimes more dense than necessary, this could be due to the page limitations. For example, Section 4 can be divided into subsections that distinguishes whether it is describing the exploration, exploitation, graph connectivity/bandit princple in PageRank. More examples are determined line-by-line in the questions section of this review.
* Lack of inclusion of code for reproducing offline link prediction results on the OGBL Datasets in supplementary material. This is the largest contributing factor to the score provided in this review, further explanation about this concern is detailed in 'Concerns about PRB performance' within the Questions section of this review.

**Questions:**

* Line 135: Are there specific citations to contextual bandits literature that inspires the proposed pseudo-regret metric?
* Line 179-180: This statement is certainly interesting, but how does it relate to PRB directly?

Concerns about Clarity:
------------------------------
* Line 181-193: Do the super nodes allow PRB to extend to the node classification? From node classification, is PRB then updated to handle link prediction by considering the connections between serving nodes and super nodes? This seems to be the case, since the papers moves from the initial definition of PRB's neural bandit-style architecture onto PRB's applications within the link prediction and node classification tasks. It is difficult to tell without more context or definitions on the relation between super nodes and PRB. Additionally, this paragraph reads much like a mathematical proof, which is good to represent individual components within the system. However, a diagram on how PRB is applied to the node classification task and then how PRB is transformed for the link prediction task would provide clarity through a visual example.


Concerns about PRB performance:
----------------------------------------------
* Given how neural bandit models are applied to online scenarios, I understand that the metric of choice for testing neural-bandit methods is regret (or pseudo-regret for PRB). Is it possible to test the accuracy of PRB for offline node classification? This is not necessarily a concern, but offline node classification results could provide insight into whether PRB's performance with pseudo-regret can translate to high-levels of accuracy.
* As mentioned in Lines 242-243: the larger the input graph, the more difficult the link prediction task is for PRB. Does that mean that the more links which PRB is required to predict, the more components of the graph PRB is then required to exploit and explore? Is there another reason why neural bandit models have difficulty with larger graphs? Do these difficulties mean neural bandits suffer: when performing link prediction and node classification, with time and space complexity limiitations, or something else?
* Considering Lines 242-243, along with PRB outperforming all other tested models, the lack of a script in the current supplementary materials to test how well PRB performs with the Hits@K metric on the much larger ogbl datasets is concerning. I kindly request the authors provide a script for replicating the results detailed in Table 3 for PRB and another model such as BUDDY, along with each model's recommended hyperparameters for the ogbl datasets.

**Limitations:**

Yes.

---

> ### Author Rebuttal · Authors · 2024-08-07
>
> Thank you for your constructive feedback and precious time. Here, we try our best to address the questions and concerns in the form of Q&A. Similar questions raised by other reviewers are addressed in the Global Response. Additional content has been added to the 1-page PDF to better address reviewers' questions.
>
> ---
> ## Q1: Clarity of descriptions
>
> Thank you for your suggestion, and we will divide Section 4 into three subsections to introduce exploitation, exploration, and PageRank, respectively.
>
> ---
> ## Q2: Details for reproducing OGBL results
> We are happy to provide a script to reproduce the results of PRB in the offline link prediction setting. Please refer to our newly submitted anonymous link for additional supplementary codes and hyperparameter settings.
>
> We also provide detailed experimental set-ups for all our experiments including both online and offline settings on **Global Response Q2**.
>
> For all graph-based baselines, we used the source code and followed the same hyper-parameter settings from [1] (source code link provided in the paper). The results of the baselines are sourced from Table 2 of [1]. These baseline results are also commonly used in other works such as [2,3].
>
> ---
> ## Q3: Citations about pseudo-regret metric
> Our pseudo-regret metric is inspired by contextual bandits literature such as neural contextual bandits [4] and neural active learning [5] to evaluate sequential decision-making. The pseudo-regret metric is a widely used evaluation metric in the literature of contextual bandits [7,8].
>
> ---
> ## Q4: Relation of statement 179-180 to PRB
> As stated in Section 4, PRB incorporates a PageRank component to address Line 8 in Algorithm 1 of the manuscript. To mitigate the additional time and space complexity introduced by this component, we aim to adapt methods that can efficiently solve the PageRank problem. The statement in Lines 179-180 highlights that Line 8 in Algorithm 1 can be accelerated and solved efficiently by using a PageRank variant, such as the one proposed by [6], thereby ensuring the overall efficiency of PRB.
>
> ---
> ## Q5: Clarity for node classification algorithm
> Thanks for your valuable suggestions. We've drawn a figure with an example (**Figure 1 in Global PDF**) to illustrate the process of transforming the node classification to the link prediction problem. Then, we use the method described in Lines 187-189 to generate contexts for supernode. With this transformation, we can directly apply PRB to this problem and predict the links between serving nodes and super nodes.
>
> ---
> ## Q6: Offline node classification accuracy
> Regarding the reviewer's question, we evaluated the accuracy performance of the PRB against three bandit-based baselines in the offline node classification task.
>
> In this experiment setup, we randomly split nodes on each dataset into 60\%/20\%/20\% for training/validation/testing. We follow the same setup for each method in **Global Response Q2**.  We then evaluate the accuracy of each trained model on the testing set. The accuracy of the test set was assessed over 10 runs to ensure robustness.
>
> The results are shown in the following table. PRB demonstrates the overall best accuracy performance. This indicates that the low regret of PRB's performance can translate into high-level accuracy.
>
> |Methods|Cora| Citeseer|Pubmed|
> |--|-|-|-|
> | NeuralTS | 73.2 | 66.5     | 82.4   |
> | NeuralUCB| 75.7 | 66.7     | 83.5   |
> | EE-Net   | 77.1 | 70.8     | 84.2   |
> | **PRB**  | **82.6** | **73.5**     | **87.4**   |
>
> We also provide the related code in our newly submitted anonymous link.
>
> ---
> ## Q7: Difficulty of large graphs
> The reviewer's observation is correct. The more target candidate nodes there are, the more options are needed to exploit and explore, increasing the task's difficulty. This mirrors real-world decision-making, where having more options makes it harder to choose the best one. Importantly, our theoretical analysis shows that the cumulative regret of PRB grows *sublinearly* with the size of the candidate node pool. However, existing graph-based methods may also suffer from the increasing complexity of the graph
> despite not providing a theoretical performance upper bound for analysis.
>
> Moreover, to better show the scalability of our method, we recorded the inference time of PRB and competitive baselines in both online and offline settings.
>
> **Table 3 in Global PDF** reports the inference time (one round in seconds) of bandit-based methods on three datasets for online link prediction. Although PRB takes a slightly longer time, it remains in the same order of magnitude as the other baselines. We adopt the approximated methods from [6] for the PageRank component to significantly reduce computation costs while ensuring good empirical performance.
>
> **Table 4 in Global PDF** reports the inference time (one epoch of testing in seconds) of graph-based methods on three datasets for offline link prediction. PRB is faster than SEAL and shows competitive inference time as compared to other baselines.
>
> ---
> **References**
> [1] Neural Common Neighbor with Completion for Link Prediction. ICLR 2024 \
> [2] Graph neural networks for link prediction with subgraph sketching. ICLR 2023 \
> [3] Neo-gnns: Neighborhood overlap-aware graph neural networks for link prediction. NeurIPS 2021\
> [4] EE-Net: Exploitation-Exploration Neural Networks in Contextual Bandits. ICLR 2022\
> [5] Neural Active Learning with Performance Guarantees. NeurIPS 2021 \
> [6] Everything Evolves in Personalized PageRank. WWW 2023 \
> [7] Neural contextual bandits with UCB-based exploration. ICML 2020 \
> [8] Improved algorithms for linear stochastic bandits. NeurIPS 2011

---

> > ### Comment · Reviewer_bPwg · 2024-08-08
> >
> > Thank you to the authors for the additional clarity on descriptions, limitations, time complexity, and further experiments. The answers provided in Question 1 and 3-7 relieve my concerns related to said questions.
> >
> > As a response to the answers for Question 2, 6 and the Senior Area Chair. My apologies for asking for an updated code sample for this review, especially after the submission deadline. The intent was not to break NeurIPS policy but to ensure reproducibility of results since the original submission did not include tests to replicate the results on OGB datasets. As such, the new anonymous link will not be considered in this review, thanks to the authors for their cooperation.
> >
> > Regardless, considering the advancements PRB makes to neural-bandit architectures with measurable upper-bounds on performance and improved learning on graphs, in addition to answering all other questions. I am raising the score for this review.

---

> > > ### Author Response · Authors · 2024-08-08
> > >
> > > Dear Reviewer bPwg,
> > >
> > > Thank you so much for your feedback and professionalism in reviewing our paper. We are very glad that our response has helped alleviate your concerns. We will update the manuscript based on our discussion and publish all parts of the codes once this paper is published.
> > >
> > > Sincerely,\
> > > Authors

---

### Official Review · Reviewer_AzV9 · 2024-07-09

**Soundness:** 2
**Presentation:** 3
**Contribution:** 3
**Rating:** 5
**Confidence:** 4

**Summary:**

This paper introduces PRB (PageRank Bandits), a novel algorithm for link prediction in graph learning that combines contextual bandits with PageRank to balance exploitation and exploration. Framing link prediction as a sequential decision-making process, the paper provides a new reward formulation and theoretical performance guarantees. PRB demonstrates superior performance over traditional methods in both online and offline evaluations.

**Strengths:**

1. The integration of PageRank with contextual bandits is a compelling concept, and the motivation behind the proposed PRB algorithm is clear.
2. Regret analysis validates the PRB algorithm's efficacy, demonstrating that its cumulative regret approaches sublinear growth.
3. The PRB algorithm exhibits robust performance across real-world tasks, achieving impressive outcomes in both online and offline settings.
4. The organization of this paper is well-structured and straightforward, facilitating ease of comprehension.

**Weaknesses:**

1. The paper does not provide detailed descriptions of experimental settings, such as model hyperparameter configurations and optimizer choices.
2. The complexity analysis of the PRB algorithm, including both time and space complexity, is not thoroughly discussed.

**Questions:**

1. Please respose the above weaknesses first.
2. In practical experiments, how are the transition matrix and the parameter $\alpha$ chosen in the PRB algorithm?

**Limitations:**

The authors discussed the limitations in the Appendix.

---

> ### Author Rebuttal · Authors · 2024-08-07
>
> Thank you very much for your constructive feedback and precious time. Here, we will try our best to address the questions and concerns in the form of Q&A. Since some of your questions are also raised by other reviewers, we have moved some answers to the Global Response. We also include additional content in the 1-page PDF file of Global Response to better answer reviewers' questions.
>
> ---
>  ## Q1: Experiment setting
> Please refer to **Global Response Q2**, where we provide the setups for both online and offline link prediction experiments and report the hyperparameter settings for PRB and all baselines.
>
> ---
> ## Q2: Time and space complexity
> **Time Complexity:**
> Let $n$ be the number of nodes, $t$ be the index of the current round of link prediction, $k$ be the number of target candidate nodes, $d$ be the number of context dimensions, and $p$ be neural network width.
>
> **Online Setting:**
> Let $m_t$ be the number of edges at round $t$.  In the setting of online link prediction, the time complexity of PRB is $O(kdp + m_t k)$, where the first term is the cost of calculating the exploitation-exploration score for each candidate node and the second term is the cost of running PageRank, following [1].  The space complexity is $O(n + m_t)$ to store node weights and edges.
>
> **Offline Setting:**
> Let $m$ be the number of edges in the testing dataset. Let $F$ be the number of target links to predict.
> Then, the inference time complexity of PRB for $F$ links is $O(ndp) + \tilde{O}(mF)$. The first term is the cost of calculating the exploitation-exploration score for each node.  The second term is the cost of PageRank [1].
> The comparison with existing methods is listed in the following table:
> | Method | Complexity |
> |--------|------------|
> | SEAL   | $O( n^{l'+1} p^2 F  )$ |
> | BUDDY  | $O(n^2 p + nh +(h + p^2)F )$ |
> | NCNC   | $O(n^2d + n d^2 + (n^2d +nd^2)F )$ |
> | PRB    | $O(ndp) + \tilde{O}(mF)$ |
>
>  $l'$ is the number of hops of the subgraph in SEAL and $h$ is the complexity of hash operation in BUDDY.
>
> **Space Complexity:** The space complexity of PRB is $O(nd + m)$ to store the context vector for each node and run PageRank.
>
> | Method | Complexity |
> |--------|------------|
> | SEAL   | $O(n^{l'+1}d)$ |
> | BUDDY  | $O(d + h)$ |
> | NCNC   | $O(n^2d)$ |
> | PRB    | $O(n + m)$ |
>
> Moreover, we recorded the inference time of PRB and competitive baselines in both online and offline settings.
> The following table reports the inference time (one round in seconds) of bandit-based methods on three datasets for online link prediction. Although PRB takes a slightly longer time, it remains in the same order of magnitude as the other baselines. We adopt the approximated methods from [1] for the PageRank component to significantly reduce computation costs while ensuring good empirical performance.
> | Methods   | MovieLens | GrQc  | Amazon Fashion |
> |-----------|-----------|-------|----------------|
> | NeuralUCB | 0.11      | 0.01  | 0.02           |
> | NeuralTS  | 0.10      | 0.01  | 0.02           |
> | EE-Net    | 0.17      | 0.03  | 0.04           |
> | PRB       | 0.20      | 0.03  | 0.04           |
>
> The following table reports the inference time (one epoch of testing in seconds) of graph-based methods on three datasets for offline link prediction. PRB is faster than SEAL and shows competitive inference time as compared to other baselines.
> | Methods | Cora  | Pubmed | Collab |
> |---------|-------|--------|--------|
> | SEAL    | 6.31  | 22.74  | 68.36  |
> | Neo-GNN | 0.12  | 0.24   | 9.47   |
> | Buddy   | 0.27  | 0.33   | 2.75   |
> | NCNC    | 0.04  | 0.07   | 1.58   |
> | PRB     | 0.11  | 0.58   | 3.52   |
>
> ---
>
> ## Q3: Choice of transition matrix and $\alpha$
> **Transition matrix:**
> In our experiment, the transition matrix $P$ is computed as $D^{-1}A$, where $A$ is the adjacency matrix and $D$ is the degree matrix of the graph, following existing works such as [1,2].
>
> **Choice of $\alpha$:** For our experimental implementation, we conducted a grid search for $\alpha$ over {0.1, 0.3, 0.5, 0.85, 0.9}, as shown in **Figure 2 of Global PDF**. We found that $\alpha = 0.85$ achieves the best empirical performance, so we set $\alpha = 0.85$ for PRB in all experiments.
> Additionally, we would like to point out that in existing works of PageRank [1,2,3,4,5], the decay factor $\alpha$ is typically set to 0.85, which has demonstrated great empirical success.
>
> ---
> **References**
> [1] Everything Evolves in Personalized PageRank. WWW 2023
> [2] Fast and accurate random walk with restart on dynamic graphs with guarantees. WWW 2018
> [3] Tpa: Fast, scalable, and accurate method for approximate random walk with restart on billion scale graphs. ICDE 2018
> [4] Temporal pagerank. ECML PKDD 2016
> [5] Efficient pagerank tracking in evolving networks. ACM SIGKDD 2015

---

> > ### Comment · Reviewer_AzV9 · 2024-08-13
> >
> > Thank you for the author's response. Considering the comments from other reviewers, I will maintain my score.

---

> > > ### Author Response · Authors · 2024-08-13
> > >
> > > Dear Reviewer AzV9,
> > >
> > > Thank you so much for your feedback and professionalism in reviewing our paper. We will update the manuscript based on your suggestions, which are very helpful.
> > >
> > > Sincerely,\
> > > Authors

---

### Official Review · Reviewer_EydH · 2024-07-11

**Soundness:** 2
**Presentation:** 2
**Contribution:** 2
**Rating:** 5
**Confidence:** 3

**Summary:**

This paper reformulates link prediction as a sequential decision-making process and propose a algorithm that combines contextual bandits with PageRank for collaborative exploitation and exploration. The experiments validate the effectiveness of the method.

**Strengths:**

1. The problem is interesting and the paper is organized well.
2. The experiments show the effectiveness of the method.

**Weaknesses:**

1. In the introduction, some recent works use GNN models to investigate the graph embedding in temporal networks and the embedding also evolves over time and is applicable to various downstream tasks. However, the relevant works are missing.
2. For the node classification, how to pre-determine the number of clusters?
3. For the PageRank, whether the hyperparameter $\alpha$ need to be learned, and how to determine the best value?
4. Link prediction is well studied problem, why do we really need bandits to study this problem? The proposed reason is not very convincing.

**Questions:**

See above.

**Limitations:**

See above.

---

> ### Author Rebuttal · Authors · 2024-08-07
>
> We sincerely thank the reviewer for your constructive feedback and precious time. Here, we try our best to address the questions and concerns in the form of Q&A. Since some of your questions are also raised by other reviewers, we have moved some answers to the Global Response. We also include additional content in the 1-page PDF file of the Global Response to better answer reviewers' questions.
>
> ---
> ## Q1: Related work of temporal Link prediction
> Thank you for pointing out these important relevant works. We have conducted a related literature review and included additional experiments to compare PRB with temporal graph neural networks. We appreciate it if the Reviewer could help to provide some related work that we may miss.
>
> **Related Work**.
> Representation learning on temporal graphs for link prediction has been widely studied in recent years to exploit patterns in historical sequences, particularly with GNN-based methods [1,2,3,4,5,6]. However, these approaches are still conventional supervised-learning-based methods that chronologically split the dataset into training and testing sets. Specifically, these methods train a GNN-based model on the temporal training data and then employ the trained model to predict links in the test data. In contrast, we formulate link predictions as sequential decision-making, where each link prediction is made sequentially. In each round of link prediction, after making the prediction, the learner directly receives feedback (i.e., whether the prediction is correct or not). The learner can then leverage this feedback (reward) to update the model for the next round of link prediction. Therefore, these conventional methods cannot directly apply to the setting of online link predictions that we focus on.
>
> As stated in our later reply to Q4 (please also refer to **Global Response Q1**), compared to conventional graph-based approaches for link prediction, our bandit-based method offers the following three advantages: (1) Adaptation over time; (2) Balancing exploitation and exploration; and (3) Theoretical performance guarantee.  Link prediction methods on temporal graphs may adapt their models by incorporating timestamp information, but balancing exploitation and exploration and providing a theoretical performance guarantee are two unique advantages of our method.
>
> We will include all of these discussions in our revised version to broaden the scope of our method.
>
> **Additional Experiment**.
> To further demonstrate the effectiveness of our approach, we adapted PRB to the temporal link prediction setting by training it on the training dataset and only making predictions on the test dataset. Following the same setup for temporal link prediction as in [1], we chronologically split the dataset with ratios of 70\%-15\%-15\% for training, validation, and testing. Since PRB is not designed to incorporate link features, we selected three datasets without available link features: UCI, Enron, and LastFM. The setup of PRB follows the setup described in **Global Response Q2**. We compared PRB against 10 baselines: JODIE, DyRep, TGAT, TGN, CAWN, EdgeBank, TCL, GraphMixer, DyGFormer [2], and FreeDyG [1]. Detailed descriptions of each baseline can be found in [1].
>
>
> **The results are reported in Table 2 of Global PDF**, with baseline results sourced from [1].
>
>
>
>
> PRB outperforms other temporal graph-based methods, demonstrating its unique advantage in balancing exploitation and exploration. We will include the additional conducted experiments in our paper to broaden the applicability of PRB.
>
> To reproduce the results, please find the codes in our newly submitted anonymous link.
>
> ---
> ## Q2:  Number of clusters in node classification
>
> We've drawn a figure (Please refer to **Figure 1 of Global PDF**), to illustrate the process of transforming the node classification to the link prediction problem, where we use one supernode to represent each cluster. Therefore, the number of classes will be the number of supernodes, which is prior knowledge in our problem setting.
> Then, we use the method described in Lines 187-189 to generate contexts for the supernode. With this transformation, we can directly apply PRB to this problem and predict the links between serving nodes and supernodes.
>
> ---
> ## Q3: Choice of $\alpha$
>
> In the experiments, we conducted a grid search for $\alpha$ over {0.1, 0.3, 0.5, 0.85, 0.9}, as shown in **Figure 2 of Global PDF**. We found that $\alpha = 0.85$ achieves the best empirical performance, so we set $\alpha = 0.85$ for PRB in all experiments. Additionally, we would like to point out that in existing works of PageRank [7,8,9], the decay factor $\alpha$ is typically set to 0.85, which has demonstrated great empirical success.
>
> ---
> ## Q4: Motivation of solving link predictions in the contextual bandit setting
>
> Please refer to **Global Response Q1** for detailed answers, where we elaborate on three advantages of solving link predictions via contextual bandits: (1) PRB can adapt over time by leveraging the feedback from each round; (2) PRB can balance exploitation and exploration in sequential link predictions; and (3) PRB has a theoretical performance guarantee.
>
> ---
> **Reference**
>
> [1] FreeDyG: Frequency Enhanced Continuous-Time Dynamic Graph Model for Link Prediction. ICLR 2024 \
> [2] Towards better dynamic graph learning: New architecture and unified library. NeurIPS 2023 \
> [3] Do we really need complicated model architectures for temporal networks? ICLR 2023 \
> [4] Inductive representation learning on temporal graphs. ICLR 2020 \
> [5] Inductive representation learning in temporal networks via causal anonymous walks. ICLR 2021 \
> [6] Temporal graph networks for deep learning on dynamic graphs. arXiv 2020 \
> [7] Everything Evolves in Personalized PageRank. WWW 2023 \
> [8] Fast and accurate random walk with restart on dynamic graphs with guarantees. WWW 2018\
> [9] Temporal pagerank. ECML PKDD 2016

---

> > ### Comment · Reviewer_EydH · 2024-08-13
> >
> > I have read the rebuttal where the authors partially addressed my concerns. The authors try their best to find a way to enhance the quality of the paper. I would like to reconsider my score.

---

> > > ### Author Response · Authors · 2024-08-13
> > >
> > > Dear Reviewer EydH,
> > >
> > > Thank you so much for your feedback and professionalism in reviewing our paper. We are very glad that our response has addressed some of your concerns. We will update the manuscript based on our discussion.
> > >
> > > If you have any further concerns or questions, please let us know and we would be very happy to discuss them.
> > >
> > > Sincerely,\
> > > Authors

---

### Official Review · Reviewer_616o · 2024-07-13

**Soundness:** 3
**Presentation:** 3
**Contribution:** 3
**Rating:** 6
**Confidence:** 3

**Summary:**

In light of the dynamic and evolving nature of real-world graphs, PageRank Bandits is proposed to make the prediction task consistently meet the context and adapt over time. Both experimental results and theoretical analysis are solid. But the paper organization is not so well.

**Strengths:**

--It is novel to combine contextual bandits with PageRank to find a better tradeoff between exploitation and exploration with graph connectivity.

--Both contextual bandits and link prediction models are listed in related work. Moreover, both kinds of methods are compared as baselines in the experiments.

--Comprehensive experiments are conducted to verify PageRank Bandits’ effectiveness and ablation study is also done to indicate that the model design is reasonable.

--Theoretical analysis is also provided in the paper and the appendix to show its feasibility.

**Weaknesses:**

--The advantage of solving the link prediction task in a bandit setting is unclear. As mentioned in Paragraph 2 from Line 29, the dynamic and evolving nature of real-world graphs should be captured in the link prediction model. Experimental result analysis is short. For example, the failure of graph based baselines is owing to lack of exploration in Line 339. It is hard to understand this single sentence because that the graph neural network provides the capability of message passing.

--Contextual bandits have been widely deployed in practice for online personalization and recommendation tasks.

--Experimental setting is too simple to be self-contained. Training details are missing.

--Font sizes of figures in the appendix are small.

--The ablation study of bandits and PageRank should be included in the experiments to verify its main claim that the combination of contextual bandits and PageRank to achieve the balancing between exploitation and exploration.

--The inference time should also be compared between the bandit methods and the graph based methods for link prediction in the experiments.

**Questions:**

--The inference time should also be compared between the bandit methods and the graph based methods for link prediction in the experiments.

**Limitations:**

Limitation should be provided.

---

> ### Author Rebuttal · Authors · 2024-08-07
>
> We would like to thank the reviewer for your constructive feedback and precious time. Here, we try our best to address the questions and concerns in the form of Q&A. Since some of your questions are also raised by other reviewers, we have moved some answers to the Global Response. We also include additional content in the 1-page PDF file of the Global Response to better answer reviewers' questions.
>
> ---
> ## Q1: Advantages of solving link predictions in the contextual bandit setting
> Please refer to **Global Response Q1** for detailed answers, where we elaborate on three advantages of solving link predictions via contextual bandits: (1) PRB can adapt over time by leveraging the feedback from each round; (2) PRB can balance exploitation and exploration in sequential link predictions; and (3) PRB has a theoretical performance guarantee.
>
> ---
> ## Q2: Experiment setting
> Please refer to **Global Response Q2**, where we provide the setups for both online and offline link prediction experiments and report the hyperparameter settings for PRB and all baselines.
>
> ---
> ## Q3: Ablation study of bandits and PageRank
> We've conducted the ablation study for the bandit component and PageRank component of PRB, respectively. The result is placed in Appendix A of the original manuscript (Lines 518-525, Figure 6).
>
> In Figure 6 of the original manuscript, we compare the performance of PRB with EvePPR (PageRank component) and EEnet (bandit component). On one hand, PRB significantly outperforms PageRank, because PRB can integrate the exploitation and exploration of node context in sequential link predictions to boost the performance. On the other hand, PRB surpasses Bandits, as PRB can leverage the graph's structure and connectivity through enhanced PageRank. Overall, PRB consistently achieves lower regrets compared to both PageRank and Bandits, demonstrating the effectiveness of combining exploitation-exploration trade-off with PageRank.
>
> ---
> ## Q4: Inference time comparison
> We recorded the inference time of PRB and competitive baselines in both online and offline settings.
>
> The following table reports the inference time (one round in seconds) of bandit-based methods on three datasets for online link prediction. Although PRB takes a slightly longer time, it remains in the same order of magnitude as the other baselines. We adopt the approximated methods from [1] for the PageRank component to significantly reduce computation costs while ensuring good empirical performance.
> | Methods   | MovieLens | GrQc  | Amazon Fashion |
> |-----------|-----------|-------|----------------|
> | NeuralUCB | 0.11      | 0.01  | 0.02           |
> | NeuralTS  | 0.10      | 0.01  | 0.02           |
> | EE-Net    | 0.17      | 0.03  | 0.04           |
> | PRB       | 0.20      | 0.03  | 0.04           |
>
> The following table reports the inference time (one epoch of testing in seconds) of graph-based methods on three datasets for offline link prediction. PRB is faster than SEAL and shows competitive inference time as compared to other baselines.
> | Methods | Cora  | Pubmed | Collab |
> |---------|-------|--------|--------|
> | SEAL    | 6.31  | 22.74  | 68.36  |
> | Neo-GNN | 0.12  | 0.24   | 9.47   |
> | Buddy   | 0.27  | 0.33   | 2.75   |
> | NCNC    | 0.04  | 0.07   | 1.58   |
> | PRB     | 0.11  | 0.58   | 3.52   |
>
> ---
> ## Other Suggestions
> We sincerely thank the reviewer for detailed and valuable suggestions. We will revise the manuscript by integrating more experiment settings into the experiment section, enlarging the font size of the figures in the Appendix, and emphasizing the advantages of combining PageRank with Bandits.
>
> ---
> **Reference**
> [1] Everything Evolves in Personalized PageRank. WWW 2023

---

### Author Rebuttal · Authors · 2024-08-07

---
## Q1: Advantages of solving link predictions in the contextual bandit setting
(1)  **Adaptation over Time**.
   As links in real-world graphs are typically formed sequentially, it is natural to frame link predictions as a sequential decision-making process. Each link prediction can be viewed as an individual decision, and we introduce the regret metric to formulate this goal. To minimize cumulative regret, the model must adapt over time, leveraging the rewards from each round of decision-making. In contrast, traditional supervised models are often static; they are trained on a dataset and then make predictions on a separate testing dataset without further adaptation.

(2) **Balancing Exploitation and Exploration**.
The challenge of balancing exploitation and exploration is prevalent in link predictions. The learner must exploit previously formed links to select high-confidence links, while also exploring under-explored or low-confidence links to gather information for long-term benefit. \
*Example*: Users and videos form a bipartite graph on a short-video social media platform. In round $t$, let $u_t$ be the user being served and $H_t$ be their interaction history. The goal is to select and display videos that $u_t$ is likely to "like". By exploiting $H_t$ (e.g., $u_t$ has previously liked many sports videos), the platform recommends another popular sports video (exploitation). Alternatively, the platform could explore by recommending a cooking video uploaded by a new user (exploration), which $u_t$ has not interacted with before. If $u_t$ likes the video, it reveals a new preference; if not, it provides insights into $u_t$’s dislikes and the potential quality of the new user's content. *While this exploration may not be optimal for round $t$, it offers long-term benefits by improving future link predictions*.\
Contextual bandits offer a principled approach to managing the trade-off between exploitation and exploration, which PRB can utilize. In contrast, most graph-based methods lack an explicit exploration strategy.

(3) **Theoretical Performance Guarantee**.
 Formulating link prediction within a contextual bandit setting allows us to offer a rigorous theoretical performance guarantee for PRB. This theoretical framework ensures that the cumulative regret of PRB increases sublinearly with the number of rounds in the worst case.
 In other words, with high probability, the number of wrong predictions PRB makes up to round $T$ is upper bounded by $\tilde{O}(\sqrt{T})$.
 In contrast, most graph-based methods do not provide such theoretical guarantees.

---

## Q2: Experiment Setting

**Online Link Prediction Setups**.
For all bandit-based methods including PRB, for fair comparison,  the exploitation network $f_1$ is built by a 2-layer fully connected network with 100-width. For the exploration network of EE-Net and PRB, we use a 2-layer fully connected network with 100-width as well. For NeuralUCB and NeuralTS, following the setting of [1,2], we use the exploitation network $f_1$ and conduct the grid search for the exploration parameter $\nu$ over {0.001, 0.01, 0.1, 1} and for the regularization parameter $\lambda$ over {0.01, 0.1, 1}. For the neural bandits NeuralUCB/TS, following their setting, as they have expensive computation costs to store and compute the whole gradient matrix, we use a diagonal matrix to make an approximation. For all grid-searched parameters, we choose the best of them for comparison and report the average results of 10 runs for all methods.
For all bandit-based methods, we use SGD as the optimizer for the exploitation network $f_1$.
Additionally, for EE-Net and PRB, we use the Adam optimizer for the exploration network $f_2$.
For all neural networks, we conduct the grid search for learning rate over {0.01, 0.001, 0.0005, 0.0001}.
For PRB, we strictly follow the settings in [3] to implement the PageRank component. Specifically, we set the parameter $\alpha = 0.85$ after grid search over {0.1, 0.3, 0.5, 0.85, 0.9}, and the terminated accuracy $\epsilon = 10^{-6}$.
For each dataset, we first shuffle the data and then run each network for 10,000 rounds ($t = 10,000$). We train each network every 50 rounds when $t < 2000$ and every 100 rounds when $2000 < t < 10,000$.


**Offline Link Prediction Setups**. For the graph-based methods, we strictly follow the experimental and hyperparameters settings in [4,5] to reproduce the experimental results.
Offline link prediction task requires graph links to play dual roles as both supervision labels and message passing links. For all datasets, the message-passing links at training time are equal to the supervision links, while at test and validation time,
disjoint sets of links are held out for supervision that are never seen during training.
All hyperparameters are tuned using Weights and Biases random search, exploring the search space of hidden dimensions from 64 to 512, dropout from 0 to 1, layers from 1 to 3, weight decay from 0 to 0.001, and learning rates from 0.0001 to 0.01. Hyperparameters yielding the highest validation accuracy are selected, and results are reported on a single-use test set.
For PRB, we use setups similar to those in the online setting. We utilize the exploitation network $f_1$ and exploration network $f_2$ both with 500-width. We set the training epoch to 100 and evaluate the model performance on validation and test datasets. We utilize the Adam optimizer for all baseline models. For PRB implementation, We utilize the SGD optimizer for $f_1$ and the Adam optimizer for $f_2$.

---
**References**

[1] Neural contextual bandits with UCB-based exploration. ICML 2020\
[2] Neural Thompson Sampling. ICLR 2021\
[3] Everything Evolves in Personalized PageRank. WWW 2023\
[4] Neural Common Neighbor with Completion for Link Prediction. ICLR 2024\
[5] Graph neural networks for link prediction with subgraph sketching. ICLR 2023

---

> ### Author Response · Authors · 2024-08-08
> **Anonymous Code Link in Reviewing**
>
> Dear Reviewers,
>
> We would like to take this opportunity to thank you all for your constructive feedback and detailed comments on our work. We have organized the codes into an anonymous link to support reproducing the experimental results, including:
>
> 1. The codes for reproducing results in offline link prediction for PRB and all baselines.
> 2. The codes for reproducing results in node classification accuracy.
> 3. The codes for reproducing results in *temporal* link predictions.
>
> As external links are not allowed to be directly shown on Rebuttal according to NeurIPS policy, we have submitted the anonymous code link to the Area Chair (AC) for anonymous review.  **The code link will be visible to all reviewers once we receive approval from the AC**.
>
> Thank you for your understanding, and we appreciate the AC's time and efforts in reviewing.
>
> We would also like to provide a short summary of the uploaded Global PDF, which includes:
>
> 1. [Figure 1] illustrates the process of transforming node classification to link prediction.
> 2. [Figure 2] shows the ablation study for $\alpha$.
> 3. [Table 1] provides the node classification accuracy in the offline setting.
> 4. [Table 2] reports the average precision for temporal link prediction.
> 5. [Tables 3-4] show the inference time of PRB in both online and offline settings.
> 6. [Tables 5-6] provide the time and space complexity of PRB.
>
> Thank you once again, and please let us know if you have any questions.
>
> Best regards,
> The Authors

---

> > ### Comment · Senior_Area_Chairs · 2024-08-08
> > **No external code links allowed**
> >
> > Hello Authors and Reviewers of 7007
> >
> > This is the Senior AC. I am afraid NeurIPS policies do not allow for sharing of code and other such supplementary material after the original submission deadline. Reviewers should be able to reach an opinion on the paper without the need to look at code, and likewise authors need to be able to substantiate their claims without the need to point to code.
> >
> > For the policy, see here:
> >
> > https://neurips.cc/Conferences/2024/PaperInformation/NeurIPS-FAQ
> >
> > and specifically this paragraph therein:
> >
> > "Can I submit supplementary material after the deadline? No. In 2024, technical appendices that support the paper with additional proofs, derivations, or results should be included as part of the main PDF submission. Other supplementary material such as data and code can be uploaded as a separate ZIP file before the same submission deadline."

---

### Decision · Program_Chairs · 2024-09-25

**Decision:**

Accept (poster)

**Comment:**

The authors cast the link prediction problem as a sequential decision-making process. They also propose a fusion algorithm called PageRank Bandits, which brings together contextual bandits with PageRank. The reviewers all agree that this paper is well written, and the idea of doing link prediction in the framework of contextual bandits is interesting. The authors should motivate the use of contextual bandits more. They should also clearly specify the choice of hyperparameters and experimental settings to enhance the reproducibility of the results.